# Dyna-Mind: Learning to Simulate from Experience for Better AI Agents

**Xiao Yu**[1][*], **Baolin Peng**[2][†], **Michel Galley**[2], **Hao Cheng**[2], **Qianhui Wu**[2]
**Janardhan Kulkarni**[2], **Suman Nath**[2], **Zhou Yu**[1], **Jianfeng Gao**[2]
[1]Columbia University, NY  [2]Microsoft Research, Redmond
`{xy2437, zy2461}@columbia.edu`
`{baolinpeng, jfgao}@microsoft.com`

## Abstract

Reasoning models have recently shown remarkable progress in domains such as math and coding. However, their expert-level abilities in math and coding contrast sharply with their performance in long-horizon, interactive tasks such as web navigation and computer/phone-use. Inspired by literature on human cognition, we argue that current AI agents need "vicarious trial and error"—the capacity to mentally simulate alternative futures before acting—in order to enhance their understanding and performance in complex interactive environments. We introduce Dyna-Mind, a two-stage training framework that explicitly teaches (V)LM agents to integrate such simulation into their reasoning. In stage 1, we introduce Reasoning with Simulations (ReSim), which trains the agent to generate structured reasoning traces from expanded search trees built from real experience gathered through environment interactions. ReSim thus grounds the agent's reasoning in faithful world dynamics and equips it with the ability to anticipate future states in its reasoning. In stage 2, we propose Dyna-GRPO, an online reinforcement learning method to further strengthen the agent's simulation and decision-making ability by using both outcome rewards and intermediate states as feedback from real rollouts. Experiments on two synthetic benchmarks (Sokoban and ALFWorld) and one realistic benchmark (AndroidWorld) demonstrate that (1) ReSim effectively infuses simulation ability into AI agents, and (2) Dyna-GRPO leverages outcome and interaction-level signals to learn better policies for long-horizon, planning-intensive tasks. Together, these results highlight the central role of simulation in enabling AI agents to reason, plan, and act more effectively in the ever more challenging environments.

## 1 Introduction

Recent advances in language models have unlocked impressive reasoning capabilities in domains such as mathematics and programming (Shao et al., 2024; Jimenez et al., 2024). However, many emerging applications unfold in complex environments that require multi-step reasoning, such as web navigation (Zhou et al., 2024b; Deng et al., 2023), deep research (Gou et al., 2025a; Du et al., 2025), and computer/phone-use tasks (Xie et al., 2024; Rawles et al., 2025). Success in these domains depends not only on the ability to decompose goals and reflect on past progress, but also on AI agents' ability to construct accurate world models that capture the structure and dynamics of increasingly complex environments (Shao et al., 2024; Jimenez et al., 2024).

Insights from human cognition indicate why such ability to model and simulate complex environments is critical. Neuroscience research (Tolman, 1948; Daw et al., 2005; Daw & Dayan, 2014; Bennett, 2023) highlights the emergence of the neocortex as a turning point in intelligence, enabling early mammals to engage in "vicarious trial and error": mentally simulating possible futures, evaluating their consequences, and selecting advantageous actions without directly experiencing each option. This ability greatly enhanced adaptability and decision-making, which we argue is equally essential for reasoning in long-horizon AI agent tasks.

---

[*]Work done during internship at Microsoft Research; [†]Project Lead

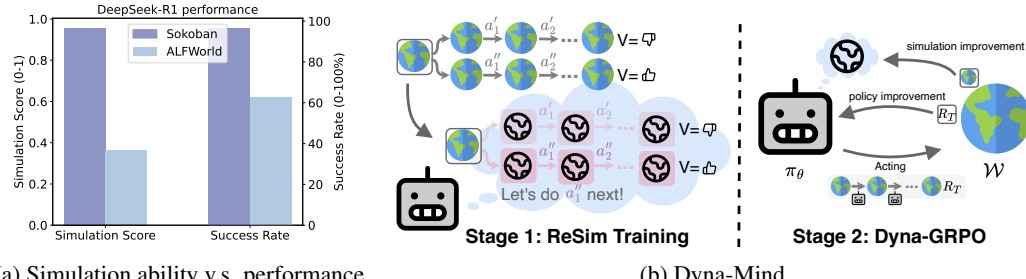

(a) Simulation ability v.s. performance        (b) Dyna-Mind

Figure 1: We find the performance of strong reasoning models is heavily affected by its ability to simulate in different environments (**left**). We introduce Dyna-Mind, a two-stage training framework to integrate and improve simulation ability of AI agents (**right**).

Empirical evidence supports this view. In Figure 1a, we observe that while strong reasoning models such as DeepSeek-R1 can simulate and solve structured environments like Sokoban, their performance drops sharply in more complex domains such as ALFWorld—both in simulation accuracy and overall task success (also see Section 4.1.2). Initial attempts to address this limitation, such as Dyna-Think (Yu et al., 2025b), integrate simulation into reasoning through distilling simplified traces and adding auxiliary next-state prediction tasks. However, these methods rely on the strong capability of reasoning models to directly generate synthetic simulation data, which can embed errors and biases.

To overcome this limitation, we present Dyna-Mind, an improved two-stage training framework to teach (V)LM agents to simulate the environment by directly learning from real experiences. In stage 1 training, we propose Reasoning with Simulations (RESIM) to algorithmically construct reasoning traces using expanded search trees obtained from real environment interactions, and train a policy model using these reasoning traces. In stage 2 training, we further improve the policy and its simulation ability using online reinforcement learning (RL). We introduce Dyna-GRPO, a novel algorithm that utilizes both outcome rewards and intermediate states from rollouts to improve the simulation ability of the policy. Extensive experiments on two widely used synthetic benchmarks (Sokoban and ALFWorld) and one realistic benchmark (AndroidWorld) show the effectiveness of each stage of the framework. Our results indicate that (1) RESIM's reasoning traces effectively teach AI agents to simulate; and (2) Dyna-GRPO, by leveraging both outcome rewards and intermediate interactions, learns better policies for long-horizon, planning-intensive tasks. These findings highlight the importance of world simulation ability for reasoning in long-horizon tasks. [1]

## 2 RELATED WORK

**(V)LM as decision making agents** The use of (visual) language models as autonomous agents has been explored in a wide range of applications such as interactive game playing (Wang et al., 2023; Feng et al., 2025), computer, phone, and browser uses (Xie et al., 2024; Zhou et al., 2024b; Rawles et al., 2025), software engineering (Jimenez et al., 2024; Yang et al., 2024), and more. Early works include reactive agents (Yao et al., 2023b) that directly prompts an (V)LM to make decisions on immediate observations without simulation or planning approaches, hindering performance on complex long-horizon tasks. Recent advances include: (1) search-based methods (Yao et al., 2023a; Zhou et al., 2024a; Koh et al., 2024; Yu et al., 2023; 2025a) that augments (V)LM agents with algorithms such as BFS, DFS, and MCTS; and (2) hierarchical, multi-agent methods (Zheng et al., 2024; Agashe et al., 2024; 2025; Liu et al., 2025; Gou et al., 2025b) that orchestrate multiple specialized agents to complete long-horizon tasks. While these methods show improvements, they often introduce substantial overheads during inference, such as requiring additional interactions with the environments or designing complex heuristics to orchestrate multiple agents. We focus on enhancing a single (V)LM agent by integrating simulation into its reasoning via training.

**Training (V)LM agents** Early methods in training (V)LM agents mostly rely on supervised learning (SFT) with human annotations or data synthesized by state-of-the-art (reasoning) models (Zeng et al., 2023; Chen et al., 2024; Zhang et al., 2024; Xu et al., 2025). Recently, many methods such as Feng et al. (2025); Wang et al. (2025b); Wei et al. (2025a;b) leverage reinforcement learning

---

[1]Code available at `https://github.com/jasonyux/Dyna-Mind`

(RL) with verifiable rewards to directly train agents to complete tasks by *prompting* them to reason before taking actions, following the success of DeepSeek-R1 (DeepSeek-AI et al., 2025a). However, it remains unclear whether extensive reasoning is necessary for all scenarios (Shojaee et al., 2025), and what aspects of such reasoning is essential for long-horizon tasks (Yu et al., 2025b). In this work, we specialize in integrating and improving the simulation ability of (V)LM agents during reasoning, and show that planning with world simulation is crucial for long-horizon tasks.

**World models and Dyna algorithms**  Beyond task completion, real-world interaction data contains rich information that can be used to help decision making. Early examples include Dyna algorithms (Sutton, 1991), which combine model-based and model-free methods to efficiently learn optimal policies. Given a set of real-world rollout data, Dyna (1) separately train a world model using these rollouts; (2) perform additional simulated rollouts with the world model; and (3) update the policy using both real and simulated rollouts. Applications of world model training have been explored in work such as Chae et al. (2025); Gu et al. (2025), facilitating search algorithms such as MCTS to improve performance; and applications of Dyna include Deep Dyna-Q (Peng et al., 2018), Switch-DDQ (Wu et al., 2018), and more (Zou et al., 2020; Yu et al., 2025b). However, these approaches either result in modular systems (a separate policy and world model) or require accessing state-of-the-art reasoning models (e.g., DeepSeek-R1). Our work does not rely on strong reasoning models, and focuses on integrating and improving simulation as part of an agent's reasoning process.

## 3  DYNA-MIND

Research in human cognition (Daw et al., 2005; Daw & Dayan, 2014; Bennett, 2023) as well as in games like chess, go, and othello (Schrittwieser et al., 2020; Li et al., 2024; Nanda et al., 2023; Chae et al., 2025) suggests that strong agents implicitly store and use a (compressed) representation of the world to enhance their decision-making. This perspective highlights two key questions in existing approaches to improve (V)LM agents for long-horizon tasks: (1) how to synergize world simulations with reasoning; and (2) how to improve the simulation ability to help improve the policy.

To address these questions, we introduce Dyna-Mind, a two-stage training framework to teach (V)LM agents to plan with simulations during their reasoning and improve their task performance. We detail these two training stages next in Section 3.2 and Section 3.3, respectively.

### 3.1  NOTATION

Completing tasks in complex, realistic environments is typically formulated as a Markov Decision Process of $(\mathcal{S}, \mathcal{A}, \mathcal{T}, \mathcal{R})$. In the generic setting of multi-step tasks, an agent $\pi_\theta$ receives an instruction and observation[2] from the environment $s_t \sim \mathcal{S}$ at time step $t$, generates an action $a_t \sim \pi_\theta(\cdot|s_t)$, and transitions to the next state $s_{t+1} \sim \mathcal{T}(s_t, a_t)$. This process is repeated until the task is completed or until reaching a maximum number of steps, upon which a terminal reward $r_T \sim \mathcal{R}(s_T, a_T)$ is provided based on whether the task is completed successfully or not. In the context of simple text games such as Sokoban Schrader (2018), a state $s_t$ can represent the complete game state, and an action $a_t$ is one of "left", "right", "up", "down" (after some reasoning process). In more complex environments such as AndroidWorld (Rawles et al., 2025), a state $s_t$ is the current screenshot of the android device, and an action $a_t$ can be "tapping on a coordinate (x,y)", "swiping up", "swiping down", etc. We note that since we aim to train agents to generate simulations *within* their reasoning process, any text that represents simulation is always part of the response $a_t$.[3] Any variant of the symbol $s$ represents *real* states from environment interactions, unless explicitly stated otherwise.

### 3.2  REASONING WITH SIMULATIONS (RESIM)

To enable an agent to simulate during its reasoning, we first construct imitation learning data where the reasoning process consists of explicitly planning with simulations. Different from prior work

---

[2]Technically, any input to the agent from our environments is an observation (as in POMDP) instead of a state. However, to simplify notation we used $s$ to generally denote the agent's input from the environment.

[3]As action plan/final action are always extracted from model response, we use $a$ (by slight abuse of notation) to denote either the full response or the extracted executable action. Distinctions are made clear in context. Example model response for each benchmark is provided in Table A4, Table A5, and Figure A1.

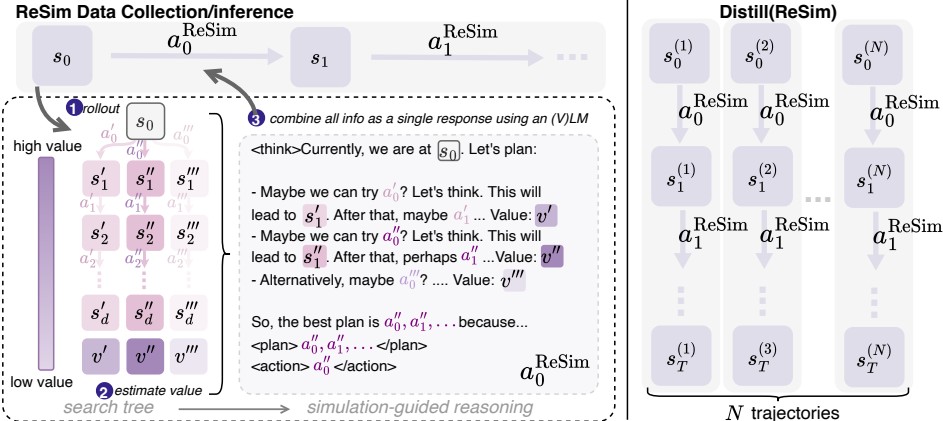

Figure 2: RESIM integrates simulation into reasoning ($a_t^{\text{ReSim}}$) by using expanded search trees built through *real* environment interactions (**left**). RESIM then trains an agent to directly generate such simulation-guided reasoning trace $a_t^{\text{ReSim}}$ without any algorithm support (**right**).

such as Yu et al. (2025b) that leverages superior LLMs such as DeepSeek-R1 which already shows world modeling capability in its reasoning traces (see Section 4.1.1 for more details), we construct simulation-guided reasoning traces using search trees built from *real environment interactions*.

**RESIM Data Collection** To construct reasoning data with rich simulations, we leverage algorithms such as depth first search (DFS) to construct search trees based on environment interactions, and then use an (V)LM to aggregate *the entire search tree* into a single reasoning response $a^{\text{ReSim}}$ for later training. Specifically, given a state $s$, RESIM first uses a rollout model $\pi_\theta$ to generate $b$ rollouts from $s$ up to depth $d$. This rollout model can be a specialized/finetuned LLM (see Section 4.1) or simply prompting a generic LLM (see Section 4.2). Then, RESIM uses a value function $V_\nu$ to provide an estimate of the quality of each of the partial rollouts, where the $V_\nu$ can be implemented as either a finetuned value model (see Section 4.1) or using LLM-as-a-judge (see Section 4.2). Finally, we use a generic (V)LM to aggregate all these rollouts and their values into a single response $a^{\text{ReSim}}$ by prompting the (V)LM to 1) first independently summarize each partial rollout, *which contains ground-truth future states information from the environment*; and 2) then aggregate all these summaries into a coherent response conditioned on the current state $s$ and previous $h$ actions and states, and choose the best plan and the next immediate action for execution. The final chosen action from $a^{\text{ReSim}}$ is then executed in the environment, and this process is repeated until the task is solved or until a maximum number of steps is reached. We illustrate this process in Figure 2 Left and Algorithm 3. We note that since RESIM essentially converts real search trees into a single reasoning trace, it is not limited to (1) agent-environment interactions; (2) specific search algorithms used in this work. We believe other domains such as agent-user-environment interactions or other algorithms such as MCTS are also applicable, which we leave for future work.

**RESIM Distillation** Since each response $a^{\text{ReSim}}$ encapsulates an entire search tree in its reasoning, we directly use $a^{\text{ReSim}}$ as the training target given an input $s$ to teach the model to perform simulation-guided reasoning without any algorithm support. We illustrate this in Figure 2 Right. Specifically, given a collection of trajectories $\tau = \{s_0, a_0^{\text{ReSim}}, s_1, a_1^{\text{ReSim}}, \cdots, s_T, a_T^{\text{ReSim}}\}$ produced by RESIM inference, we use SFT to train the model to directly generate each $a_t^{\text{ReSim}}$ given the current state $s_t$ as well as a maximum history of $h$ previous actions and states in the trajectory (i.e., the same input used by other inference methods such as REACT).

## 3.3 DYNA-GRPO

While RESIM provides a principled way to synergize simulation with reasoning, it is computationally expensive and relies on multiple modules (a rollout model, a value function, and a (V)LM to aggregate the search tree into a single response) to construct training data. Additionally, such offline training may limit models' generalization ability to new tasks. To address this, we propose DYNA-GRPO, a modification of GRPO (Shao et al., 2024) to further improve the model's simulation ability during

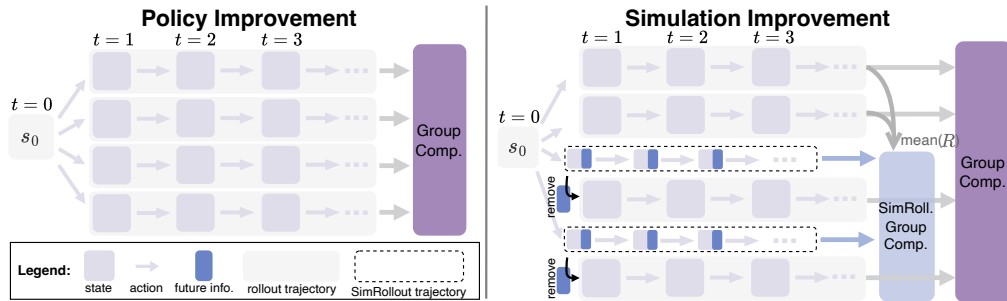

Figure 3: DYNA-GRPO iterates between policy improvement (**left**) and world model improvement (**right**), optimized by GRPO. During policy improvement, we perform grouped policy rollouts with GRPO. During simulation improvement, we perform both policy rollouts and simulation refinement rollouts (see Figure 4), and trains the model to directly generate an improved policy as well as to better perform simulation refinement when provided with future-states information.

online RL without using any search or additional modules. The standard GRPO objective $\mathcal{J}_{\mathrm{GRPO}}$ is:

$$\mathbb{E}_{\tau \sim \pi_{\theta_{\mathrm{old}}}} \left[ \frac{1}{GT} \sum_{i=1}^{G} \sum_{t=1}^{T} \min \left( \rho_\theta(a_t^{(i)}) A(a_t^{(i)}), \mathrm{clip}(\rho_\theta(a_t^{(i)}), 1 \pm \epsilon) A(a_t^{(i)}) \right) - \beta D_{\mathrm{KL}}(\pi_\theta || \pi_{\theta_{\mathrm{ref}}}) \right],$$

where $\rho_\theta(a) = \frac{\pi_\theta(a|s)}{\pi_{\theta_{\mathrm{ref}}}(a|s)}$ is the importance sampling ratio, $\beta$ is the KL regularization coefficient, and $A = A_{\mathrm{GRPO}}$ is the episode-level advantage function (Wang et al., 2025b; Feng et al., 2025):

$$A(a_t^{(i)}) = A_{\mathrm{GRPO}}(\tau^{(i)}) = \frac{R(\tau^{(i)}) - \mathrm{mean}(\{R(\tau^{(j)})\}_{j=1}^{G})}{\mathrm{std}(\{R(\tau^{(j)})\}_{j=1}^{G})}, \quad R(\tau^{(i)}) = \sum_{t=1}^{T} R(s_t, a_t),$$

where $G$ is the group size, $R(\cdot)$ is the reward provided by the environment, with $R(s_t, a_t) = -0.1$ for non-terminal steps and $R(s_T, a_T) = 10.0$ or $R(s_T, a_T) = 0.0$ for terminal steps when task succeeded or failed, respectively.

However, RL algorithms such as GRPO aim to optimize a policy only using *scalar* rewards $R_T$ but do not provide any direct training signal on refining the reasoning process or world model simulations. We propose DYNA-GRPO to address this, by *additionally* incorporating future state(s) information $s_{t+1}, s_{t+2}, \cdots$ as *textual* signals to help improve the model's response $a \sim \pi_\theta(\cdot|s_t)$ during RL training. Since textual signals cannot be directly "optimized", we propose SIMROLLOUT to instead *prompt the underlying model* to refine its simulation in $a \sim \pi_\theta(\cdot|s_t)$ utilizing real future state(s) $s_{t+1}, s_{t+2}, \cdots$ *during RL rollouts*. Then, during optimization we train the policy to both directly generate the refined action and also to improve its "simulation refinement" ability (DYNA-GRPO). We detail these two modifications below.

**SIMROLLOUT** In simulation refinement rollout (SIMROLLOUT), at each state $s_t$ we first sample a response $a \sim \pi_\theta(\cdot|s_t)$; then extract the final chosen plan $\{\hat{a}_1, \hat{a}_2, \cdots, \hat{a}_d\}$ up to depth $d$ from $a$ and execute them in the environment to obtain ground truth next-states $\{s'_{t+1}, s'_{t+2}, \cdots, s'_{t+d}\}$; and finally prompt $\pi_\theta$ again to refine its response $a$ given these real future states $a^{\mathrm{refine}} \sim \pi_\theta(\cdot|s_t^{\mathrm{refine}}), s_t^{\mathrm{refine}} \equiv \{s_t \oplus a \oplus s'_{t+1} \oplus \hat{a}_2 \oplus \cdots \oplus s'_{t+d}\}.$[4] We illustrate this rollout process in Figure 4 and provide the pseudo-code in Algorithm 2. We note that this is different from methods such

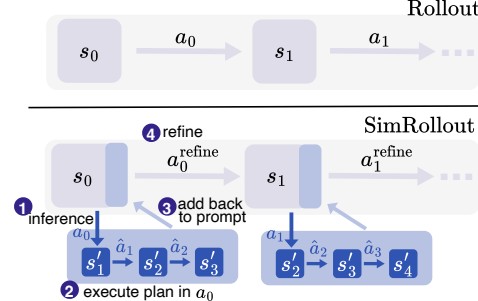

Figure 4: SIMROLLOUT generates refined action per state $s_t$ using real environment interactions

as Reflexion (Shinn et al., 2023), which performs reflection at the end of the episode utilizing success/failure information, and is also not intended for any training purposes. Empirically, we find the resulting $a^{\mathrm{refine}}$ indeed improves the policy's simulation and performance (see Appendix D.4).

---

[4]Since $d$ is determined dynamically by the model, we capped it at 5 to ensure training stability.

**DYNA-GRPO Training**  To utilize refined trajectories from SIMROLLOUT during RL, we follow Dyna algorithms to improve the model's policy and simulation ability iteratively. Specifically, DYNA-GRPO iterates between (1) *simulation improvement* where models learn from refined policies that use future states information from SIMROLLOUT to improve its simulation ability; and (2) direct *policy improvement* where models are trained on standard rollouts without future-state access, allowing it to better integrate simulation ability into decision-making. We illustrate both training processes in Figure 3, and detail the overall algorithm in Algorithm 1.

---

**Algorithm 1** DYNA-GRPO

**Require:** policy $\pi_\theta$, environment $\mathcal{T}$, group size $G$
**Require:** hyperparameters $G, N, n_\mathcal{T}, n_\pi$
1: **for** $N$ training iterations **do**
2:     *// simulation improvement*
3:     **for** $n_\mathcal{T}$ steps **do**
4:         *// see Algorithm 2*
5:         $\{\tau'\}, \{\tau'_{\text{refine}}\} \leftarrow \text{SimRollout}(\pi_\theta, \mathcal{T}, G/2)$
6:         $\{\tau\} \leftarrow \text{Rollout}(\pi_\theta, \mathcal{T}, G/2)$
7:         Update $\pi_\theta$ with $\text{GRPO}(\{\tau\} \cup \{\tau'\})$
8:         Update $\pi_\theta$ with $\text{GRPO}(\{\tau'_{\text{refine}}\})$ using $A_{\text{refine}}$
9:     **end for**
10:    *// policy improvement*
11:    **for** $n_\pi$ steps **do**
12:       $\{\tau\} \leftarrow \text{Rollout}(\pi_\theta, \mathcal{T}, G)$
13:       Update $\pi_\theta$ with $\text{GRPO}(\{\tau\})$
14:    **end for**
15: **end for**
16: **return** $\pi_\theta$

---

During simulation improvement, for each task we (1) first perform SIMROLLOUT with a group size of $G/2$, collecting refined trajectories with and without future-state information removed: $\tau' = \{s_0, a_0^{\text{refine}}, s_1, a_1^{\text{refine}}, \cdots\}$ and $\tau'_{\text{refine}} = \{s_0^{\text{refine}}, a_0^{\text{refine}}, s_1^{\text{refine}}, a_1^{\text{refine}}, \cdots\}$; (2) then perform standard rollouts with group size of $G/2$; (3) combine these standard rollouts $\tau$ with refined trajectories $\tau'$ into a single group of size $G$ and perform GRPO on this combined group; (4) finally utilize $\tau'_{\text{refine}}$ to also improve the model's simulation refinement ability, using the following modified advantage to reward refinements that both correctly solves the task *and* improves upon (the mean reward of) standard policy rollouts which does not access future states:

$$A_{\text{refine}}(\tau_{\text{refine}}^{(i)}) = \begin{cases} 1.0, & \text{if } \tau_{\text{refine}}^{(i)} \text{ is correct and } R(\tau_{\text{refine}}^{(i)}) > \max(\bar{R}, \bar{R}^{\text{refine}}) \\ 0.0, & \text{otherwise} \end{cases},$$

where $\bar{R} = \frac{1}{G/2} \sum_{i=1}^{G/2} R(\tau^{(i)})$ is the mean reward of the standard policy rollouts (line 6 of Algorithm 1); $\bar{R}^{\text{refine}} = \frac{1}{G/2} \sum_{i=1}^{G/2} R(\tau_{\text{refine}}^{(i)})$ is mean reward from SIMROLLOUT (line 5 of Algorithm 1). During policy improvement, we perform standard policy rollouts without future state information, optimized by GRPO using episode-level advantage (Feng et al., 2025; Wang et al., 2025b).

## 4  EXPERIMENTS

We first evaluate Dyna-Mind on two "synthetic" environments (Sokoban and ALFWorld) that require efficient planning for successful task completion. These lightweight environments allow us to provide detailed analysis of the different reasoning styles as well as different RL algorithms. Then, we extend our methods to a more complex and realistic environment (AndroidWorld).

### 4.1  TEXT GAMES

**Benchmarks**  Sokoban (Schrader, 2018) is a grid-world game where the agent needs to push boxes to target destinations while avoiding obstacles, and successful task completion requires spatial planning to avoid deadlock situations. ALFWorld (Shridhar et al., 2021) is a text-based embodied environment where the agent needs to locate/interact with objects to complete household tasks using natural language instructions. To evaluate the agent's generalization ability, we construct training set, an in-distribution (ID) test set, and an out-of-distribution (OOD) test set. For Sokoban, we use training set with 6x6 room layouts with 1 box and 1 destination; ID test set with different 6x6 room layouts than training; and OOD test set with 8x8 room layouts with 1 box and 1 destination. For ALFWorld, we directly use the official training, ID, and OOD test splits from Shridhar et al. (2021).

**Baselines setup**  To evaluate RESIM, we compare against (1) ReACT based prompting methods with models such as GPT-4o (OpenAI, 2024), Claude-3.7 (Anthropic, 2025), DeepSeek-V3 (DeepSeek-AI et al., 2025b), and DeepSeek-R1 (DeepSeek-AI et al., 2025a); and (2) training methods that distill the reasoning traces from strong policy models such as DeepSeek-R1. To evaluate stage 2 DYNA-GRPO training, we compare against other popular group-based RL algorithms such as RLOO (Kool et al.,

Table 1: Performance on text game environments such as Sokoban and ALFWorld. "Gen. Token" denotes the average number of tokens generated per turn relative to that of Qwen2.5-7B-Instruct. All training in stage-1 and stage-2 are based on Qwen2.5-7B-Instruct. All results are averaged over 3 runs. Our methods are highlighted in gray.

| Method | Gen. Token | Sokoban | | | ALFWorld | | |
|---|---|---|---|---|---|---|---|
| | | ID | OOD | AVG | ID | OOD | AVG |
| REACT(Qwen2.5-7B-Instruct) | 1.0x | $25.8_{\pm1.8}$ | - | - | $35.4_{\pm1.9}$ | - | - |
| REACT(Qwen2.5-32B-Instruct) | 2.7x | $36.7_{\pm4.2}$ | - | - | $36.2_{\pm3.3}$ | - | - |
| REACT(GPT-4o) | 1.5x | $37.8_{\pm1.0}$ | - | - | $51.3_{\pm2.1}$ | - | - |
| REACT(Claude-3.7-Sonnet) | 2.3x | $70.3_{\pm1.2}$ | - | - | $46.1_{\pm1.0}$ | - | - |
| REACT(DeepSeek-V3) | 2.5x | $57.0_{\pm1.6}$ | - | - | $55.2_{\pm1.0}$ | - | - |
| REACT(DeepSeek-R1) | 14.5x | $\mathbf{96.6}_{\pm0.2}$ | - | - | $62.5_{\pm0.5}$ | - | - |
| RESIM | 2.0x | $96.4_{\pm0.2}$ | - | - | $\mathbf{87.7}_{\pm1.1}$ | - | - |
| *Dyna-Think* | | | | | | | |
| DIT(R1)+DDT($\hat{\mathcal{T}}$) | 24.2x | $74.0_{\pm1.4}$ | $57.5_{\pm1.2}$ | $65.8_{\pm1.9}$ | $63.2_{\pm1.5}$ | $56.7_{\pm2.8}$ | $58.9_{\pm2.3}$ |
| *Dyna-Mind Stage 1 (SFT)* | | | | | | | |
| DISTILL(V3) | 2.1x | $49.2_{\pm1.1}$ | $34.4_{\pm1.3}$ | $41.8_{\pm1.1}$ | $58.9_{\pm1.1}$ | $56.7_{\pm1.0}$ | $57.8_{\pm1.2}$ |
| DISTILL(R1) | 24.0x | $\mathbf{72.5}_{\pm2.9}$ | $\mathbf{57.0}_{\pm1.9}$ | $\mathbf{64.8}_{\pm2.5}$ | $59.4_{\pm1.5}$ | $54.2_{\pm3.9}$ | $56.8_{\pm3.5}$ |
| DISTILL(RESIM) | 2.0x | $71.9_{\pm1.5}$ | $55.5_{\pm1.6}$ | $63.7_{\pm1.9}$ | $\mathbf{78.9}_{\pm2.1}$ | $\mathbf{69.3}_{\pm1.3}$ | $\mathbf{74.1}_{\pm1.8}$ |
| *Dyna-Mind Stage 2 (RL)* | | | | | | | |
| DISTILL(RESIM) + RLOO | 2.2x | $78.1_{\pm1.8}$ | $65.1_{\pm1.3}$ | $71.3_{\pm0.9}$ | $85.9_{\pm1.3}$ | $85.4_{\pm2.0}$ | $85.5_{\pm2.0}$ |
| DISTILL(RESIM) + GRPO | 2.1x | $79.1_{\pm1.3}$ | $67.8_{\pm0.6}$ | $73.1_{\pm1.4}$ | $87.0_{\pm3.2}$ | $87.1_{\pm1.1}$ | $87.0_{\pm1.8}$ |
| DISTILL(RESIM) + DYNA-GRPO | 1.9x | $\mathbf{82.5}_{\pm1.5}$ | $\mathbf{70.1}_{\pm1.6}$ | $\mathbf{77.1}_{\pm1.7}$ | $\mathbf{92.5}_{\pm0.8}$ | $\mathbf{89.1}_{\pm1.3}$ | $\mathbf{90.8}_{\pm0.9}$ |

2019) and GRPO (Shao et al., 2024). Overall, we also compare against Dyna-Think (Yu et al., 2025b), which similarly uses two-stage training (DIT and DDT) to improve model's simulation ability.

**Dyna-Mind setup** To instantiate RESIM, we use Qwen2.5-32B-Instruct (Qwen et al., 2025) as rollout and value function models, finetuned on rollouts obtained by using DeepSeek-V3 (see Appendix D.3 for more details) and use DeepSeek-V3 as the LLM to aggregate the search tree into a single response. For Sokoban, we use $d = 5, b = 16, b_{\text{train}} = 2$; for ALFWorld, we use $d = 2, b = 24, b_{\text{train}} = 4$. We note that all models used by RESIM are by themselves much weaker than other models such as DeepSeek-R1 as well as RESIM itself. Since DeepSeek-R1 and RESIM have a higher success rate than DeepSeek-V3, to isolate improvement from better reasoning from simply training with more (diverse) data, we thus *only used trajectories where all methods correctly solved the task* for stage 1 training. This results in a total of 207 trajectories in Sokoban and 200 trajectories in ALFWorld from each method (DeepSeek-R1, DeepSeek-V3, and RESIM) in the subsequent stage 1 training.

To instantiate DYNA-GRPO, we continue training the best model from stage 1 distillation. To ensure a fair comparison, we use identical hyperparameters for all methods (RLOO, GRPO, and DYNA-GRPO), when applicable. For DYNA-GRPO, we use $n_{\mathcal{T}} = 10$ and $n_{\pi} = 10$ for Sokoban and $n_{\mathcal{T}} = 10$ and $n_{\pi} = 20$ for ALFWorld. For more setup details, please see Appendix D.5.

#### 4.1.1 MAIN RESULTS

In the upper section of Table 1, we first evaluate RESIM's performance against other strong reasoning models such as DeepSeek-R1. Then, we compare different training methods to integrate/improve the simulation ability of the policy model. In Table 1, we first find that RESIM achieves near-perfect performance on Sokoban (96.4% success) and a strong performance on ALFWorld (87.7% success), significantly outperforming all other methods. On Sokoban, we find strong reasoning models such as DeepSeek-R1 also achieves near-perfect performance, which we attribute to R1's ability to correctly simulate Sokoban game states (but not on ALFWorld) during its reasoning process (see Section 4.1.2 for empirical results). In contrast, RESIM utilizes ground-truth simulations from search trees, and hence was able to achieve strong performance in both environments.

In stage 1 training, we find DISTILL(RESIM) achieves a similar performance to DISTILL(R1) on Sokoban but significantly outperforms both DISTILL(V3) and DISTILL(R1) on ALFWorld. Additionally, since RESIM constructs reasoning traces consists almost entirely of only planning via simulation (see Figure 2 Left), DISTILL(RESIM) outputs *11x less tokens* on average compared to DISTILL(R1).

Table 2: Measuring simulation ability of different models across different training stages. We report the average success rate and the simulation ability (Sim Score $\in [0, 1]$) averaged across all trajectories. We also report the correlation coefficient $r$ between the success rate and the simulation score.

| Method | Sokoban | | ALFWorld | |
|---|---|---|---|---|
| | Success | Sim Score | Success | Sim Score |
| REACT(Qwen2.5-7B-Instruct) | $25.8_{\pm1.8}$ | $0.21_{(r\,=0.64)}$ | $35.4_{\pm1.9}$ | $0.18_{(r\,=0.46)}$ |
| REACT(DeepSeek-V3) | $57.0_{\pm1.6}$ | $0.54_{(r\,=0.81)}$ | $55.2_{\pm1.0}$ | $0.35_{(r\,=0.68)}$ |
| REACT(DeepSeek-R1) | $\mathbf{96.6}_{\pm0.2}$ | $0.93_{(r\,=0.96)}$ | $62.5_{\pm0.5}$ | $0.36_{(r\,=0.70)}$ |
| RESIM | $96.4_{\pm0.2}$ | $\mathbf{1.00}_{(-)}$ | $\mathbf{87.7}_{\pm1.1}$ | $\mathbf{1.00}_{(-)}$ |
| *Dyna-Think* | | | | |
| DIT(R1)+DDT($\hat{\mathcal{T}}$) | $74.0_{\pm1.4}$ | $0.62_{(r\,=0.74)}$ | $63.2_{\pm1.5}$ | $0.36_{(r\,=0.76)}$ |
| *Dyna-Mind Stage 1 (SFT)* | | | | |
| DISTILL(R1) | $\mathbf{72.5}_{\pm2.9}$ | $0.61_{(r\,=0.75)}$ | $59.4_{\pm1.5}$ | $0.34_{(r\,=0.77)}$ |
| DISTILL(RESIM) | $71.9_{\pm1.5}$ | $\mathbf{0.62}_{(r\,=0.78)}$ | $\mathbf{78.9}_{\pm2.1}$ | $\mathbf{0.37}_{(r\,=0.74)}$ |
| *Dyna-Mind Stage 2 (RL)* | | | | |
| DISTILL(RESIM) + GRPO | $79.1_{\pm1.3}$ | $0.62_{(r\,=0.65)}$ | $87.0_{\pm3.2}$ | $0.38_{(\rho\,=0.48)}$ |
| DISTILL(RESIM) + DYNA-GRPO | $\mathbf{82.5}_{\pm1.5}$ | $\mathbf{0.67}_{(r\,=0.64)}$ | $\mathbf{92.5}_{\pm0.8}$ | $\mathbf{0.43}_{(r\,=0.55)}$ |

These results indicate that that strong performance from RESIM can be learned by SFT, and that the ability to model and simulate the environment is crucial for long-horizon, planning-intensive tasks.

In stage 2 training, we continue from the best model (DISTILL(RESIM)) with online RL. In Table 1, we find that DYNA-GRPO improves upon both GRPO, RLOO, as well as Dyna-Think, while maintaining a similar output token length compared to its base model DISTILL(RESIM). This indicates that DYNA-GRPO is effective at improving the model's simulation ability during online RL training (also see Section 4.1.2 for empirical results), and that improving such simulation ability helps improve task performance. For a more detailed ablation study comparing the effect of search algorithm used by RESIM as well as DYNA-GRPO, please see Table A1 and Appendix D.1.

### 4.1.2 MEASURING SIMULATION ABILITY

Dyna-Mind aims to integrate and improve the simulation ability of agents. To measure this simulation ability, we evaluate the **Simulation Score** (Sim Score) of different models and the **Spearman Correlation Coefficient** ($r_s$) between sim score and success rate. Given a state $s_t$ and generated response $a_t \sim \pi_\theta(\cdot|s_t)$, we evaluate the simulation score of $a_t$ by 1) first prompting an LLM to extract the final action plan $(\hat{a}_1, \hat{a}_2, \cdots, \hat{a}_d)$ and the natural language description (i.e., simulation) of the corresponding imagined next-states $(\hat{s}_{t+1}, \hat{s}_{t+2}, \cdots, \hat{s}_{t+d})$ from the response $a_t$; 2) then execute the action plan in the environment to obtain ground truth next-states $\{s_{t+1}, s_{t+2}, \cdots, s_{t+d}\}$; 3) finally, prompt an LLM to judge (Zheng et al., 2023) the correctness of these simulated next-states $\hat{s}_i$ by comparing them against the ground truth $s_i$, returning a score $\in [0, 1]$. Finally, we averaged the score for each turn to obtain an overall simulation score for the trajectory. To ensure a fair judgment, we used a different LLM from all of our experiments (Qwen3-235B-A22B-Instruct (Qwen Team, 2025)). For judgment prompts, please see Appendix D.6.

We present the results in Table 2. In Table 2, we find that 1) RESIM maintains its strong success rates across both Sokoban and ALFWorld due to its perfect simulation ability (by construction), whereas DeepSeek-R1 struggled in ALFWorld as it struggles to model the environment layout; and 2) both DISTILL(RESIM) and DYNA-GRPO improve the simulation ability alongside task performance compared to their baselines. These results show that our methods helped improve the simulation ability of the model beyond simply improving task performance.

### 4.2 ANDROIDWORLD

Next, we extend our Dyna-Mind to AndroidWorld (Rawles et al., 2025) - a highly challenging benchmark that evaluates the agent's ability control and complete tasks on a virtual Android device.

**Benchmarks** AndroidWorld (Rawles et al., 2025) provides a fully functional Android environment that requires the agent to interact with Android's GUI to complete tasks across 20 real-world Android apps. Since tasks in AndroidWorld are parameterized by task types (116), we construct a training

Table 3: Performance on AndroidWorld. All training are based on Qwen2.5-VL-7B/32B-Instruct. We exclude Dyna-Think since (most) VLMs cannot predict *images*, as required by DDT($\hat{\mathcal{T}}$) training. "Gen. Token" denotes the average number of tokens generated per turn relative to that of Qwen2.5-7B-Instruct. All results are averaged over 3 runs. Our methods are highlighted in gray.

| Method | Gen. Token | AndroidWorld | | |
|---|---|---|---|---|
| | | ID | OOD | AVG |
| REACT(GPT-4o) | 1.0x | $5.1_{\pm 0.2}$ | - | - |
| REACT(Qwen2.5-VL-7B-Instruct) | 1.0x | $5.3_{\pm 0.2}$ | - | - |
| REACT(Qwen2.5-VL-72B-Instruct) | 1.1x | $19.5_{\pm 0.4}$ | - | - |
| RESIM | 2.1x | $\mathbf{34.4}_{\pm 0.4}$ | - | - |
| *Dyna-Mind Stage 1 (SFT)* | | | | |
| DISTILL-7B(Qwen2.5-VL-72B-Instruct) | 1.0x | $13.1_{\pm 0.4}$ | $8.6_{\pm 0.2}$ | $10.8_{\pm 0.6}$ |
| DISTILL-7B(RESIM) | 2.1x | $21.1_{\pm 0.4}$ | $10.2_{\pm 0.6}$ | $15.7_{\pm 0.8}$ |
| DISTILL-32B(RESIM) | 2.0x | $\mathbf{32.8}_{\pm 0.4}$ | $\mathbf{15.6}_{\pm 0.7}$ | $\mathbf{24.2}_{\pm 0.6}$ |
| *Dyna-Mind Stage 2 (RL)* | | | | |
| DISTILL-32B(RESIM) + GRPO | 2.1x | $35.3_{\pm 0.4}$ | $20.3_{\pm 0.6}$ | $27.8_{\pm 0.4}$ |
| DISTILL-32B(RESIM) + DYNA-GRPO | 1.9x | $\mathbf{40.7}_{\pm 1.0}$ | $\mathbf{22.9}_{\pm 1.0}$ | $\mathbf{31.8}_{\pm 1.0}$ |

set with 81 task types with in total 1946 tasks, an ID test set with 128 different tasks from the same task types, and an OOD test set with 128 tasks from the remaining 35 held-out task types. We use a maximum number of 15 steps and the screenshot-only modality as input. We provide an example task and action in Appendix E.1.

**Baselines setup**   Since our methods consider end-to-end training, we compare against models that are capable of directly generating executable actions given an GUI screenshot, and exclude modular systems such as Gou et al. (2025b); Agashe et al. (2025). We thus mainly compare against (1) REACT based prompting method with Qwen2.5-VL-72B/7B (Bai et al., 2025), and GPT-4o; and (2) distillation from Qwen2.5-VL-72B[5]. To evaluate stage 2 DYNA-GRPO, we compare against GRPO following Section 4.1. We exclude comparison against Dyna-Think in this experiment, because DDT($\hat{\mathcal{T}}$) trains the model to predict next-state (in this case, screenshot images), which cannot be implemented using most VLMs as they can only generate text.

**Dyna-Mind setup**   Since AndroidWorld is a highly challenging and compute-intensive environment (each episode on average takes 15-20 minutes to complete), we do not perform any rollout/value function training for RESIM. Instead, we directly prompt Qwen2.5-VL-72B as the rollout model, prompt GPT-4o as a judge to approximate the value function, and also use GPT-4o as the VLM to aggregate the rollouts into a single response in RESIM. We use $d = 1$, $b = 16$, $b_{\text{train}} = 4$ for RESIM, and a total of 128 trajectories for distillation/stage 1 training. To instantiate DYNA-GRPO, we generally followed the same recipe as Section 4.1, but used less training steps (60) as AndroidWorld is highly compute-intensive and time-consuming. For more details, please see Appendix E.2.

### 4.2.1   MAIN RESULTS

**Results**   We present the results in Table 3. In general, we observe similar results compared to Section 4.1.1. First, we find that RESIM inference significantly improves performance, and that the improved performance can be transferred to Qwen2.5-VL-7B and 32B via DISTILL(RESIM). Next, in both training stages of Dyna-Mind, we find improved performance in both ID and OOD test sets compared to baselines, including Qwen2.5-VL-72B and even RESIM. These results highlight the effectiveness of our method to improve agent's performance in complex environments.

**Error Analysis**   Compared to synthetic text games (Section 4.1.1) where RESIM achieves near-perfect performance, we find RESIM struggles in AndroidWorld despite improvements compared to baselines. After analyzing trajectories produced by RESIM, we find performance is bottlenecked by the rollout model (Qwen2.5-VL-72B), mainly due to: (1) incomplete understanding of some GUI

---

[5]We were unable to reproduce the reported performance of more recent GUI models such as UI-Tars1.5 (Qin et al., 2025), and hence focus on using Qwen2.5-VL for simplicity. Please see Appendix E.3 for more details.

interfaces and certain button functions, and (2) inability to recover after making multiple mistakes. We believe methods to improve the foundation model's capability could mitigate these problems (Wang et al., 2025a; Qin et al., 2025), which we leave for future work.

## 5 CONCLUSION

In this work, we propose Dyna-Mind to synergize reasoning with simulations for autonomous AI agents. We empirically show that an agent's ability to model and simulate the environment strongly correlates with its ability to correctly reason and complete long-horizon, planning-intensive tasks. We introduce Dyna-Mind, a two-stage training method to explicitly teach (V)LM agents to integrate and improve such simulation a part of their reasoning. In stage 1 training, we propose RESIM to train a model to simulate future states by learning to predict an expanded search tree in their reasoning. In stage 2 training, we propose DYNA-GRPO to further refine the agent's reasoning and simulation ability using online RL. Empirical results on three benchmarks show that (1) RESIM effectively teaches AI agents to simulate; and (2) DYNA-GRPO, by leveraging both outcome rewards and intermediate interactions, learns better policies for long-horizon, planning-intensive tasks.

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

## A    LLM USAGE

This work used LLMs as general-purpose writing assistants to improve the grammar and clarity of the paper. We *did not* use LLMs to generate any research ideas, automate experiments, or analyze results.

## B    ETHICS STATEMENT

Generally, while most methods and models are not designed for unethical usage, there is often potential for abuse in their applications. Autonomous AI agents can be used for a variety of tasks such as automating information gathering, software development, computer/phone-use and more. In this work, we proposed our Dyna-Mind framework to enhance the simulation ability and hence performance of AI agents. However, since AI agents are fundamentally task-agnostic, it is possible to use them for unethical tasks such as scamming or disseminating false information on the internet. We believe developing guardrails such as safety filters (OpenAI, 2022; Inan et al., 2023) are highly valuable for AI agent research. We do not condone the Dyna-Mind or its constituent methods for any unlawful or morally unjust purposes.

## C    ADDITIONAL ALGORITHMIC DETAILS

In Algorithm 2, we provide the pseudo-code for SIMROLLOUT. On a high level, SIMROLLOUT aims to generate a refined response at a given state $s_t$ with better simulation content compared to that of the original response. Specifically, SIMROLLOUT first performs normal inference $a_t \sim \pi_\theta(\cdot|s_t)$ to generate a response; extracts the plan $(\hat{a}_1, \hat{a}_2, \cdots, \hat{a}_d)$ from $a_t$ using the "<plan></plan>" tags (see Table A4 for example response with such tags); executes the extracted plan in the environment and obtain the actual next-states $\{s_{t+1}, s_{t+2}, \cdots, s_{t+d}\}$; and finally, prompts an LLM to refine the original response based on the actual next-states, using the prompt in Table A3. The resulting refined response $a_t^{\text{refine}}$ is then used as the next action $a_t$, and this process is repeated until the task is completed or a maximum number of steps is reached.

---

**Algorithm 2** Simulation Refinement Rollout (SIMROLLOUT)

---

**Require:** policy $\pi_\theta$, environment $\mathcal{T}$, group size $G$
1: repeat the following $G$ times:
2: $\quad \tau' \leftarrow \{\}, \tau'_{\text{refine}} \leftarrow \{\}, t = 0, s_0 \leftarrow T$
3: **while** not done and $t < t_{\max}$ **do**
4: $\quad\quad a \leftarrow \pi_\theta(s_t)$
5: $\quad\quad \{\hat{a}_1, \cdots, \hat{a}_n\} \leftarrow \text{extract\_plan}(a)$
6: $\quad\quad$ *// improve action a using next-state information*
7: $\quad\quad \{s_{t+1}, \cdots, s_{t+n}\} \leftarrow \{\mathcal{T}(s_t, \hat{a}_1), \cdots, \mathcal{T}(s_{t+n-1}, \hat{a}_n)\}$
8: $\quad\quad s_t^{\text{refine}} \leftarrow \text{refinement prompt}(a|s_t, a, \{s_{t+1}, \hat{a}_1, \cdots, s_{t+n}\})$ *// see Table A3*
9: $\quad\quad a^{\text{refine}} \leftarrow \pi_\theta(s_t^{\text{refine}})$
10: $\quad\quad$ *// update episode buffer*
11: $\quad\quad \tau' \leftarrow \tau' \cup \{s_t, a^{\text{refine}}\}$ *// learn improved policy*
12: $\quad\quad \tau'_{\text{refine}} \leftarrow \tau'_{\text{refine}} \cup \{s_t^{\text{refine}}, a^{\text{refine}}\}$ *// learn to refine simulations*
13: $\quad\quad s_{t+1} \leftarrow \mathcal{T}(s_t, a^{\text{refine}})$
14: $\quad\quad t \leftarrow t + 1$
15: **end while**
16: **return** $\tau', \tau'_{\text{refine}}$

---

## D    ADDITIONAL DETAILS ON TEXT GAMES

### D.1    MORE ABLATION STUDIES

In Table A1, we provide a more detailed ablation study of Dyna-Mind for Sokoban and ALFWorld. For the second-stage DYNA-GRPO training, we consider replacing $A_{\text{refine}}$ with the standard GRPO advantage (denoted as "- $A_{\text{refine}}$") and removing DYNA-GRPO (denoted as "- DYNA-GRPO"). We

note that removing DYNA-GRPO reduces Algorithm 1 to standard GRPO. For the first-stage RESIM, we consider replacing DFS with BFS (denoted as "- DFS+BFS") as well as removing DFS entirely (denoted as "- DFS"). We note that removing DFS entirely reduces DISTILL(RESIM) to simply DISTILL(V3).

## D.2 EXAMPLE TASKS AND ACTIONS

Sokoban (Schrader, 2018) is a grid-world game where the agent needs to push boxes to their destinations while avoiding obstacles. Valid actions in Sokoban are up, down, left, and right. As an example, we provide an example input state and generated action in Table A4. ALFWorld (Shridhar et al., 2021) is a text-based embodied environment where the agent needs to locate/interact with objects to complete embodied household tasks using natural language instructions. Valid actions in ALFWorld are dependent on what's available in the current state. We provide an example input state and generated action in Table A5.

Table A1: Ablation study on Dyna-Mind

| Method | Sokoban | ALFWorld |
|---|---|---|
| Dyna-GRPO | **82.5%** | **92.5%** |
| $- A_{\text{refine}}$ | 80.8% | 89.2% |
| $-$ SimRollout | 79.1% | 87.0% |
| DISTILL(RESIM) | 71.9% | 78.9% |
| $-$ DFS+BFS | 71.1% | 78.9% |
| $-$ DFS | 49.2% | 58.9% |

## D.3 RESIM IMPLEMENTATION DETAILS

We provide a pseudo-code for RESIM in Algorithm 3. For text games, we finetune Qwen2.5-32B-Instruct as rollout and value function models using DeepSeek-V3's rollouts. Specifically, we first use DeepSeek-V3 to generate 256 rollouts using tasks from the training set. Then, to train the rollout model, we simply perform SFT training on one correct rollout for each task. To train the value function, we use the trained policy model to generate the same 256 rollouts, repeated over 3 times, and compute $V(s_t)$ as the probability of successfully completing the task from $s_t$ across all trajectories that contains $s_t$, discounted by the number of remaining steps needed in the current trajectory:

$$V(s_t) = \gamma^{t_{\max}-t}\frac{1}{|\mathrm{T}|}\sum_{\tau \in \mathrm{T}} \mathbb{1}[\tau \text{ is successful}], \quad \text{where } \mathrm{T} \equiv \{\tau_1, \tau_2, \cdots | s_t \in \tau_i\}$$

where $\gamma$ is the discount factor and $t_{\max}$ is the maximum number of steps in a trajectory. In both environments, we used $\gamma = 0.95$. Finally, we finetune a separate Qwen2.5-32B-Instruct as the value function by adding a linear value head to the model architecture, and perform MSE loss training on the computed $V(s_t)$ across all states from all trajectories.

Since Sokoban and ALFWorld environments are fast, these rollouts were completed within 1 hour. For complex environments such as AndroidWorld, we directly prompt pretrained VLMs such as Qwen2.5-VL-72B and GPT-4o as rollout and value function models (Section 4.2).

## D.4 SIMULATION REFINEMENT PERFORMANCE

To empirically show that (V)LMs are capable of leveraging next-state information to improve their action, we evaluate the performance of SIMROLLOUT compared to direct prompting (REACT). We report the result in Table A2.

In general, we find that 1) all models showed improved task success rate when provided with next-state information; and 2) stronger models such as GPT-4o and GPT-4.1 (OpenAI, 2024; 2025) shows larger improvement compared to weaker models such as Qwen2.5-7B-Instruct. We believe this is because correcting its own mistakes is requires non-trivial reasoning ability, which is more difficult for weaker models such as Qwen2.5-7B-Instruct to achieve. Overall, this result indicates that world modeling error (e.g., especially for tasks such as ALFWorld) remains a significant bottleneck for (V)LM agents reasoning ability in long-horizon tasks.

## D.5 ADDITIONAL TRAINING DETAILS

To instantiate DYNA-GRPO, we continue training the best model from stage 1 distillation. To ensure a fair comparison, we use identical hyperparameters for all methods (RLOO, GRPO, and DYNA-GRPO), when applicable. We use a batch size of 8 tasks per batch, group size of $G = 8$, learning

Table A2: SIMROLLOUT performance on Sokoban and ALFWorld. We show that when provided with ground-truth next-state information (SIMROLLOUT), models *achieve better performance* compared to direct prompting (REACT).

| Base Model | Method | Sokoban | ALFWorld |
|---|---|---|---|
| Qwen2.5-7B-Instruct | REACT | $25.8_{\pm1.8}$ | $35.4_{\pm1.9}$ |
| | SIMROLLOUT | $30.0_{\pm1.4}$ | $39.1_{\pm1.6}$ |
| GPT-4o-2024-11-20 | REACT | $37.8_{\pm1.0}$ | $51.3_{\pm2.1}$ |
| | SIMROLLOUT | $41.4_{\pm1.2}$ | $64.8_{\pm2.5}$ |
| GPT-4.1 | REACT | $67.9_{\pm1.0}$ | $54.4_{\pm2.1}$ |
| | SIMROLLOUT | $71.1_{\pm1.3}$ | $67.9_{\pm2.0}$ |

---

**Algorithm 3** RESIM

---

**Require:** policy $\pi_\theta$, value function $V_\nu$, environment $\mathcal{T}$, (V)LM $M$
**Require:** hyperparameters $b, d, t_{\max}, b_{\text{train}}$
1: $\tau \leftarrow \{\}, t = 0, s_0 \leftarrow T$
2: **while** not done and $t < t_{\max}$ **do**
3:     $\{\tau^i\}_{i=1}^b \leftarrow$ sample $b$ rollouts using $\pi_\theta$ starting from $s_t$ for max $d$ steps
4:     $\{\tau^i\}_{i=1}^{b'} \leftarrow$ deduplicate $\{\tau^i\}_{i=1}^b$
5:     $\{v^i\}_{i=1}^{b'} \leftarrow$ estimate value $\{V_\nu(s_{t+d}^i)\}_{i=1}^b$
6:     *// subsample rollouts*
7:     $\tau^* \leftarrow \tau^{\arg\max_i v^i}$
8:     $\{\tau^i\}_{i=1}^{b_{\text{train}}} \leftarrow \{\tau^*\} \cup$ subsample $b_{\text{train}} - 1$ rollouts from the rest of $\{\tau^i\}_{i=1}^{b'}$
9:     *// aggregate rollouts into a single reasoning response*
10:    $\{\text{plan}^i\}_{i=1}^{b_{\text{train}}} \leftarrow$ summarize $\{M(\tau^i, v^i)\}_{i=1}^{b_{\text{train}}}$
11:    $a^{\text{RESIM}} \leftarrow$ aggregate $M(s_t, \{\text{plan}^i\}_{i=1}^{b_{\text{train}}})$
12:    *// next step*
13:    $s_{t+1} \leftarrow \mathcal{T}(s_t, a^{\text{RESIM}})$
14:    $\tau \leftarrow \tau \cup \{s_t, a^{\text{RESIM}}\}$
15:    $t \leftarrow t + 1$
16: **end while**
17: **return** $\tau$

---

rate of 1e-6, and 300 training steps in total for both Sokoban and ALFWorld. For DYNA-GRPO, we use $n_{\mathcal{T}} = 10$ and $n_\pi = 10$ for Sokoban and $n_{\mathcal{T}} = 10$ and $n_\pi = 20$ for ALFWorld. All training are performed on top of Qwen2.5-7B (Qwen et al., 2025) using 8xH100.

### D.6 SIMULATION SCORE PROMPTS

To evaluate the simulation ability of a model $\pi_\theta$, we use LLM-as-a-judge (Zheng et al., 2023) to measure the correctness and quality of the simulation generated by $\pi_\theta$ at each turn in a given trajectory. Specifically, for each $a_t \sim \pi_\theta(\cdot|s_t)$, we first prompt an LLM to extract the final action plan $(\hat{a}_1, \hat{a}_2, \cdots, \hat{a}_d)$ from $a_t$ and the corresponding natural language description of the next-states $(\hat{s}_{t+1}, \hat{s}_{t+2}, \cdots, \hat{s}_{t+d})$ from the response $a_t$. We present the prompts used for Sokoban and ALF-World in Tables A6 and A8, respectively. Then, we execute the action plan in the environment to obtain ground truth next-states $\{s_{t+1}, s_{t+2}, \cdots, s_{t+d}\}$. Finally, we prompt an LLM to judge the quality of the plan by comparing "imagined" next-states generated by $\pi_\theta$ against the ground truth next-states, using prompts in Tables A7 and A9. This results in a score $\in [0, 1]$ for each turn in the trajectory, which is then averaged across all turns to obtain an overall simulation score for the entire trajectory.

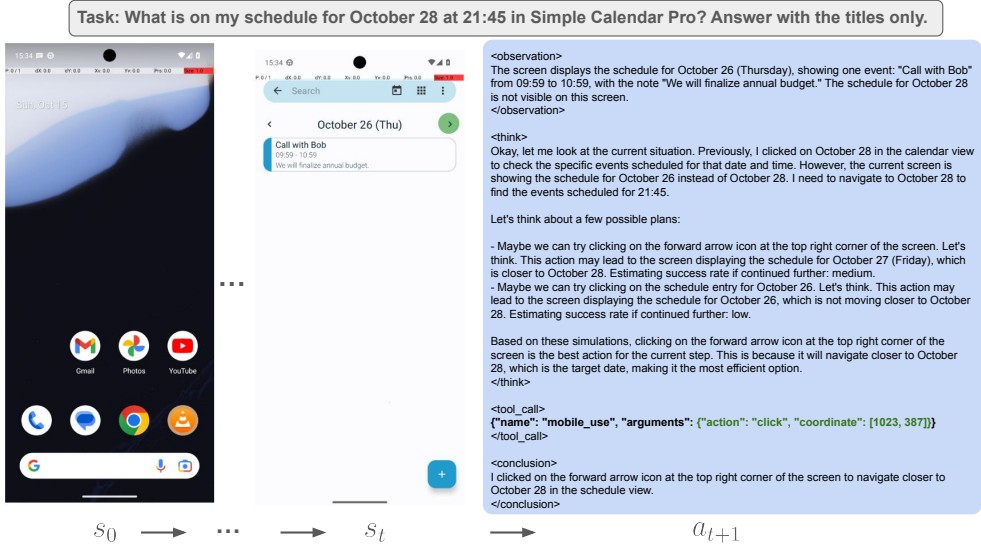

Figure A1: Example task, input screenshot, and output generated by model trained using Dyna-Mind. For clarity, we directly rendered the proposed action in $a_{t+1}$ (click at 1023,387) in green on $s_t$.

# E   ADDITIONAL DETAILS ON ANDROIDWORLD

## E.1   EXAMPLE TASK AND ACTIONS IN ANDROIDWORLD

In this work, we use the dockerized environment provided by AndroidWorld to evaluate and train all methods. We use the screenshot-only modality. In Figure A1, we present an example task, input screenshot $s_t$ from AndroidWorld, as well as an example output $a_t$ generated by models trained using Dyna-Mind. For more details on AndroidWorld, please refer to Rawles et al. (2025).

## E.2   ADDITIONAL TRAINING DETAILS

To standardize training and evaluation, we use the dockerized version of AndroidWorld and adapt the action space provided by Rawles et al. (2025).

To instantiate DYNA-GRPO, we continue training the best model from stage 1 distillation. We followed Section 4.1 and used a batch size of 8 tasks per batch, group size of $G = 8$, learning rate of 1e-6. Since AndroidWorld is highly compute-intensive and time-consuming to run, we perform a total of 60 training steps for RL training, using $n_{\mathcal{T}} = 2$ and $n_{\pi} = 8$. All training are performed on top of Qwen2.5-VL-7B-Instruct and Qwen2.5-VL-32B-Instruct (Bai et al., 2025) using 16xH100, denoted as "DISTILL-7B" and "DISTILL-32B" in Table 3, respectively.

## E.3   OTHER IMPLEMENTATION/EVALUATION DETAILS

In this work, we focus on end-to-end training (SFT + RL), and hence selected VLMs capable of directly interacting with android's GUI interface. This include models such as Qwen2.5-VL (Bai et al., 2025) and UI-Tars (Qin et al., 2025). While these models have undergone specific finetuning on mobile control tasks, at the time of the work we were unable to find evaluation scripts that supports using these models on AndroidWorld. To our best effort, we utilized the official mobile-use prompts provided by the respective repositories, as well as prompts from recent work such as (Gou et al., 2025b). However, we were unable to fully reproduce the reported performance, especially for UI-Tars 1.5. At the time of this work, we find similar concerns has also been raised publicly (e.g., `https://github.com/bytedance/UI-TARS/issues/83`, `https://github.com/UI-Tars/UI-Tars/issues/155`, `https://github.com/UI-Tars/UI-Tars/issues/121`). To this end, we focus on using Qwen2.5-VL for consistency with other experiments conducted in the rest of the paper.

Table A3: Prompt used by SIMROLLOUT to refine the agent's original response given actual next-state information. The next-state information is obtained by 1) extracting the final chosen plan from the agent's response (e.g., left, left, up in Sokoban), and 2) executing the plan in the environment to obtain the actual next states.

---

**Prompt**

---

*// ...omitting some text*
# Current observation
{current_observation}

# Example response and feedback
To help you reason and plan better, we have explored some plans for the current step and obtained the following feedback from the environment:
## Example response
{agent_original_response}
## Ground truth feedback
{actual_next_observations_after_executing_agent's_plan}

# Back to the current step
Now, the environment has been reset back to the current observation/current step. It's your turn to refine the example response based on the ground truth feedback. You should think about:
- Correctness: is the example response aligned with the feedback? did the feedback reveal some incorrect/ineffective actions in the example response?
- Progress: did the the environment feedback show positive progress towards solving the task? Note: the example response may hallucinate incorrect outcomes different from the ground truth feedback. You should avoid coming up with similar hallucinations in your response.

If you think the example response is correct and has made progress, no revision is needed and your should **directly output the example response verbatim**.
Otherwise, you should modify the example response's thinking process/plan/action to be consistent with the environment feedback. Specifically, you should:
1. **Incorporate all relevant details from the feedback** into the example response and then **improve its accuracy and progress**. Be detailed when adding information from the feedback into the response.
2. The final refined response should **read as a standalone answer, without saying its a revision or referencing the feedback section**. The output should look like the final authoritative response that faithfully incorporated relevant details from the feedback.

Now, refine the example response. Keep the same tone and format as the example response. Ensure the final plan/action is logically consistent with your (refined) thinking process. DO NOT generate anything else.

---

Table A4: Example state in Sokoban and our model's generated action

| Type | Content |
|---|---|
| Input | *// ...omitting some text about sokoban game rules*
# Current Step
Your current observation is:
Col 0 Col 1 Col 2 Col 3 Col 4 Col 5
Row 0 # # # # # #
Row 1 # _ _ # # #
Row 2 # _ P # # #
Row 3 # _ X _ _ #
Row 4 # _ _ O _ #
Row 5 # # # # # #

Currently, the player is at (2, 2); boxes are at (3, 2); unsolved targets are at (4, 3). Next to the player, there is a floor above the player at (1, 2), a box below the player at (3, 2), a floor left of the player at (2, 1), a wall right of the player at (2, 3). Your admissible actions are ["up", "down", "left", "right"].

Now it's your turn to make a move (choose ONE action only for the current step). You should first reason step-by-step about the current situation — observe the positions of boxes and targets, plan a path to push a box toward a target, and avoid traps like corners or walls. This reasoning process MUST be enclosed within \<think\> \</think\> tags.
Once you've finished your reasoning, you should choose a plan as well as an admissible action for current step. The plan needs to be presented as a sequence of admissible actions within \<plan\> \</plan\> tags (e.g., \<plan\>up, right, up\</plan\>). The action needs to be presented within \<action\> \</action\> tags (e.g., \<action\>up\</action\>). |
| Output | \<think\>Okay, let me look at the current situation. The player is at (2, 2), with a box at (3, 2) and an unsolved target at (4, 3). Let's think about a few possible plans.

Maybe we can try moving "down". This action will lead to the state where the player is at (3, 2), pushing the box to (4, 2). Is this a good next step? Let's think:
- After "down", maybe do "left". This results in the player moving to (3, 1), with the box remaining at (4, 2) and the target at (4, 3). After that, maybe "down", which leads to the player moving to (4, 1), with the box still at (4, 2) and the target at (4, 3). The player now has a box to the right and a floor above. Discounted success rate if continued further: around 80%.
- After "down", maybe do "left". This results in the player moving to (3, 1) with the box still at (4, 2). After that, maybe "down", which leads to the player moving to (4, 1) with the box and target unchanged. Next, try "right", which moves the player to (4, 2) and pushes the box onto the target at (4, 3), solving it. Discounted success rate if continued further: around 100%.

Based on these simulations, "down" is the best action for the current step. This is because the second plan (down -> left -> down -> right) achieves a discounted success rate of 100% in 4 steps.
\</think\>

\<plan\>down, left, down, right\</plan\>
\<action\>down\</action\> |

Table A5: Example state in ALFWorld and our model's generated action

| Type | Content |
|---|---|
| Input | *// ...omitting some text about alfworld rules*
Your task is to: put some book on sidetable.
Your admissable actions of the current situation are: ['go to bed 1' 'go to diningtable 1' 'go to drawer 1' 'go to drawer 2' 'go to drawer 3' 'go to drawer 4' 'go to garbagecan 1' 'go to sidetable 1' 'go to sidetable 2' 'inventory' 'look'].

Now it's your turn to take an action.
You should first reason step-by-step about the current situation, and think about plans to explore/solve the task. To plan well, you should think about the plan's future outcomes and whether it is making progress or not. This reasoning process MUST be enclosed within <think> </think> tags.
Once you've finished your reasoning, you should choose a plan as well as an admissible action for current step. The plan needs to be presented as a sequence of admissible actions within <plan> </plan> tags (e.g., <plan>go to desk 1, take mug 1 from desk 1</plan>). The action needs to be presented within <action> </action> tags (e.g., <action>go to desk 1</action>). |
| Output | <think>Okay, let me look at the current situation. Currently, I am in the middle of a room with several objects around me, including two sidetables (sidetable 1 and sidetable 2), where my task is to put a book on one of them. Let's think about a few possible plans.

Maybe we can try Go to diningtable 1. This action may lead to arriving at diningtable 1, where I see a creditcard 1, a book 3, a cdplayer 2, a laptop, a pencil, three pens, a nutshell, and a vase. Is this a good next step? Let's think:
- After Go to diningtable 1, maybe I can take book 3. This results in picking up the book 3 from the diningtable. Discounted success rate if continued further: around 80%.
- After Go to diningtable 1, maybe I can inventory myself. This may reveal that nothing is being carried, so taking anything is impossible. Discounted success rate if continued further: around 10%.

Alternatively, maybe we can try Open drawer 1. This action may lead to opening drawer 1 and seeing nothing inside. Is this a good next step? Let's think:
- After Open drawer 1, maybe "examine drawer 2" reveals a combination lock and a keychain inside. Discounted success rate if continued further: around 40%.

Based on these simulations, going to the diningtable has the highest discounted success rate (80%) as book 3 may be found. This achieves the highest success if followed by "take book 3", achieving a 80% success rate.
</think>

<plan>go to diningtable 1, take book 3</plan>
<action>go to diningtable 1</action> |

Table A6: Sokoban prompt to extract plan and imagined observation from an agent's response

| Prompt |
| --- |
| *// ...omitting some text about sokoban game rules*
# Extraction/parsing rules
Your task is to parse the response and extract the following information, IF present.
1) simulation branches
- definition: one sequence of actions the agent planned to solve the puzzle
- example: *// ...omitting some text*  2) discounted success rates
- definition: the (discounted) success rate of the simulation branch, or some numeric evaluation of how much progress that branch makes towards the goal.
- example: *// ...omitting some text*
3) final chosen branch
- definition: the simulation branch/plan that caused the agent's final decision for the current step.
- example: Based on these simulations, "up" is the best action for the current step. This is because after "up", the player can proceed with "left" and "up" again, which achieves a discounted success rate of around 90% in 3 steps.
- example output: ["up", "left", "up"]
- note: The agent chose "up" as the next action. However, we need to find the ENTIRE branch/plan that caused the agent's current decision, which is ["up", "right", "down"] in this case.
- note: if the agent did not explicitly mention which branch is chosen, you should choose the branch in the response with the highest discounted success rate.
4) final imagined observation
- definition: the imagined observation after executing the final chosen branch.
- example: After "up", "left", "up", the player pushed the box to (4,4). Now, the player is at (4, 3), with the box on target below at (4, 4). The player has a floor above at (2, 4)... The target is ... This is the best branch according to the discounted success rate. So the next action should be "up".
- example output: The player pushed the box to (4,4). Now, the player is at (4, 3), with the box on target below at (4, 4). The player has a floor above at (2, 4)... The target is ...
- note: DO NOT include the action sequence in this field. Only keep the description of the player/boxes/targets/walls position AFTER the last action in the final chosen branch.
- note: *// ...omitting some text*

# Your task
Your task is to output a JSON object in the following format:
\<json>
{
"extracted_branches": [ ...*// ...omitting some text*  ],
"extracted_final_chosen_branch": {
"actions": ["action 1", "action 2", ..., "action n"], # the ENTIRE branch/plan that caused the agent's current decision
"last_observation": "detailed, comprehensive description of the imagined observation AFTER executing the entire action sequence above.",
"discounted_success_rate": ...(a number between 0 to 100. -1 if the agent did not mention the discounted success rate)
}
}
\</json>

# Input response
{input_agent_response}

# Your task
Now, parse the response and output the JSON object enclosed by \<json> and \</json> tags. DO NOT generate anything else. |

Table A7: Sokoban prompt to evaluate the quality of the next-states imagined by an agent in its reasoning process, using the actual next-states as references.

| Prompt |
| --- |
| *// ...omitting some text about sokoban game rules*
# Evaluation rules
Provide an overall score between 0.0 and 1.0 based on the following two dimensions. Start with a score of 0.0, and add points to the score if the criteria are satisfied. Add 0.0 if a criteria is not satified. DO NOT deduct points if a criteria is not satified.
1) correctness (max 0.3 points. if exceeds 0.3, cap it at 0.3)
- in the imagination description, the coordinates of the player are correct; add 0.1 point
- in the imagination description, some of the mentioned boxes and targets have correct coordinates; add 0.05 point
- in the imagination description, all mentioned boxes and targets have correct coordinates; add 0.1 point
- in the imagination description, all mentioned walls and empty spaces have correct coordinates; add 0.05 point
2) progress (max 0.7 points. if exceeds 0.7, cap it at 0.7)
- in the reference observation, if the task is completely solved (all boxes are on targets); add 0.7 point
- relative to the current observation, if the reference observation shows major progress (unsolved boxes are moved much closer to targets, task close to be solved); add 0.5 point
- relative to the current observation, if the reference observation shows minor progress (unsolved boxes are moved a bit closer to targets); add 0.1-0.3 point, depending on how much progress is shown
- relative to the current observation, if the reference observation shows no meaningful progress; assign 0.0 point for this dimension
- in the reference observation, if the task is no longer solvable (e.g., one of the boxes is pushed into a corner and cannot be moved anymore); assign 0.0 point for this dimension
*// ...omitting some text*

# Your output format
Your task is to output a JSON object in the following format:
\<json>
{
"correctness analysis": "which correctness criteria in the evaluation rules are satisfied, and which are not.", # no more than 50 words
"correctness score": 0.0-0.3, # score for the correctness dimension
"progress analysis": "which progress criteria in the evaluation rules are satisfied, and which are not.", # no more than 50 words
"progress score": 0.0-0.7, # score for the progress dimension
"score": 0.0-1.0 # total score; add the correctness score and progress score
}
\</json>

# Current observation
{current_obs}
# Agent imagined observation after some actions
{agent_imagined_next_actions_and_obs}
# Reference observation after some actions
{actual_next_obs}

# Your task
Now, provide an evaluation analysis and score according to the evaluation rules above. Output the JSON object enclosed by \<json> and \</json> tags. DO NOT generate anything else. |

Table A8: ALFWorld prompt to extract plan and imagined observation from an agent's response

| Prompt |
| --- |

*// ...omitting some text about sokoban game rules*
# Extraction/parsing rules
Your task is to parse the response and extract the following information, IF present. *// ...omitting some text*
3) final chosen branch
- definition: the simulation branch/plan that caused the agent's final decision for the current step.
- example: Based on these simulations, "go to countertop 1" is the best action for the current step. This is because this followed by "go to countertop 2" leads to a high chance of finding a mug. Therefore, the next action for the current step should be "go to countertop 1".
- example output: ["go to countertop 1", "go to countertop 2"]
- note: The agent chose "go to countertop 1" as the next action. However, we need to find the ENTIRE branch/plan that caused the agent's current decision, which is ["go to countertop 1", "go to countertop 2"] in this case.
- note: if the agent did not explicitly mention which branch is chosen, you should choose the branch in the response with the highest discounted success rate.
4) final imagined observation
- definition: the imagined observation after executing the final chosen branch.
- example: After "go to shelf 1", "take pencil 2 from shelf 1" results in successfully picking up a pencil. This is the best branch according to the discounted success rate. So the next action should be "go to shelf 1".
- example output: The agent successfully picks up a pencil.
- note: DO NOT include the action sequence in this field. Only keep the description of the imagined observation AFTER the last action in the final chosen branch.
- note: In general, you should gather the most comprehensive and detailed description found in the response (i.e., especially try to include any mention of what objects is present). If this description is scattered across multiple places in the response, MERGE them into a single, continuous description.

# Your task
Your task is to output a JSON object in the following format: <json>
{
"extracted_branches": [ ...*// ...omitting some text*  ],
"extracted_final_chosen_branch": {
"actions": ["action 1", "action 2", ..., "action n"], # the ENTIRE branch/plan that caused the agent's current decision
"last_observation": "detailed, comprehensive description of the imagined observation AFTER executing the entire action sequence above.",
"discounted_success_rate": ...(a number between 0 to 100. -1 if the agent did not mention the discounted success rate)
} }
</json>

# Input response
{input_agent_response}

# Your task
Now, parse the response and output the JSON object enclosed by <json> and </json> tags. DO NOT generate anything else.

Table A9: ALFWorld prompt to evaluate the quality of the next-states imagined by an agent in its reasoning process, using the actual next-states as references.

| Prompt |
| --- |

*// ...omitting some text about sokoban game rules*
# Evaluation rules
Provide an overall score between 0.0 and 1.0 based on the following two dimensions.
1) correctness (max 0.3 points. if exceeds 0.3, cap it at 0.3)
- in the imagined observation, it is near identical to the reference observation; add 0.3 point
- in the imagined observation, key object(s) required by the goal are found, and they are also present in the reference observation; add 0.2 point
- in the imagined observation, relevant location(s) required by the goal are visited, and the description is somewhat aligned with the reference observation; add 0.1-0.2 point, depending on how much the description is aligned with the reference observation.
- in the imagined observation, key object(s) required by the goal are found, but these key object(s) are *NOT* present in the reference observation; assign 0.0 point
- in the reference observation, it shows nothing happened; directly assign 0.0 point for this dimension
2) progress (max 0.7 points. if exceeds 0.7, cap it at 0.7)
- in the reference observation, if the goal is completely solved (all required items are found/moved/heated/etc to or at the correct location, goal is achieved); add 0.7 point
- relative to the current observation and action history, if the reference observation shows major progress (i.e., objects required by the goal are found); add 0.5 point
- relative to the current observation and action history, if the reference observation shows minor progress (i.e., objects related to the goal are found, or locations relevant to the goal are visited); add 0.1-0.3 point, depending on *how useful this information is, beyond what was already known in the current state and action history*.
- relative to the current observation and action history, if the reference observation shows no meaningful progress (nothing happened); assign 0.0 point for this dimension
*// ...omitting some text*

# Your output format
Your task is to output a JSON object in the following format: <json>
{
"correctness analysis": "...", # no more than 50 words
"correctness score": 0.0-0.3, # score for the correctness dimension
"progress analysis": "...", # no more than 50 words
"progress score": 0.0-0.7, # score for the progress dimension
"score": 0.0-1.0 # total score; add the correctness score and progress score
}
</json>

# Action history
The current goal is to: {task_description}
{action_history}
# Current observation
{current_obs}
# Agent imagined observation after some actions
{agent_imagined_next_actions_and_obs}
# Reference observation after some actions
{actual_next_obs}

# Your task
Now, provide an evaluation analysis and score according to the evaluation rules above. Output the JSON object enclosed by <json> and </json> tags. DO NOT generate anything else.

