# OpenReview forum: "Dyna-Mind: Learning to Simulate from Experience for Better AI Agents"
_ICLR.cc/2026/Conference — ICLR 2026 Poster_

### Official Review · Reviewer_J1B4 · 2025-10-27

**Soundness:** 3
**Presentation:** 3
**Contribution:** 2
**Rating:** 6
**Confidence:** 3

**Summary:**

This paper presents Dyna-Mind, a two-stage training framework designed to equip LLMs with the ability of ***vicarious trial and error***, i.e., the capacity to mentally simulate alternative futures before acting, in order to perform better in complex, long-horizon interactive tasks. In Stage 1, the model learns to generate structured reasoning traces from search trees built on real environment interactions, grounding its reasoning in faithful world dynamics and enabling it to anticipate future states. In Stage 2, the model is further optimized via online RL, leveraging both outcome rewards and intermediate feedback from real rollouts to enhance its simulation and decision-making abilities. Experiments on Sokoban, ALFWorld, and AndroidWorld show that RᴇSɪM effectively instills simulation capabilities, while Dyna-GRPO improves policy learning for long-horizon, planning-intensive tasks. Overall, the study underscores the pivotal role of simulation in enabling AI agents to reason, plan, and act effectively in complex environments.

**Strengths:**

1. Novel conceptual contribution: the introduction of ***vicarious trial and error*** as a cognitive-inspired mechanism for AI agents is well-motivated by human reasoning literature.


2. Clear and systematic framework: the two-stage design, i.e., RᴇSɪM + Dyna-GRPO, provides a coherent pipeline from reasoning grounding to online policy refinement.


3. Empirical validation: the experiments on both synthetic, i.e., Sokoban, ALFWorld, and realistic, i.e., AndroidWorld, benchmarks demonstrate solid and consistent performance gains.


4. Interpretability: the structured reasoning traces in RᴇSɪM make the agent’s decision process more transparent and analyzable.

**Weaknesses:**

1. High computational cost: the proposed pipeline appears quite complex, involving rollouts, value function updates, search tree aggregation, and RᴇSɪM distillation, all of which may be resource-intensive. Consequently, it remains unclear whether the method can scale efficiently to large environments or applications.


2. Ablation clarity: more detailed ablation studies would help clarify the contribution of each component, particularly the individual modules within RᴇSɪM and Dyna-GRPO, to the overall performance.

**Questions:**

About $S_{t}^{\text{refine}}$, since it consists of $d$ state-action pairs, could its length pose an issue for the LLM input? Specifically, is there a risk that the sequence could become too long for the model to handle efficiently? Is the model performance sensitive to $d$?

---

> ### Author Response · Authors · 2025-11-17
> **Response to Weaknesses**
>
> We thank reviewer J1B4 for acknowledging our “novel conceptual contribution”, “clear and systematic” method, and strong and diverse results across three benchmarks (Sokoban, ALFWorld, and AndroidWorld)!
>
> ---
>
> > High computational cost: the proposed pipeline appears quite complex, involving rollouts, value function updates, search tree aggregation, and RᴇSɪM distillation, all of which may be resource-intensive. Consequently, it remains unclear whether the method can scale efficiently to large environments or applications.
>
> We note that we demonstrated our method not only on Sokoban and ALFWorld, ***but also on AndroidWorld (see Section 4.2)*** which is a highly diverse and realistic environment about controlling android devices. We believe our strong results on all three benchmarks shows the effectiveness of our method.
>
> Regarding computation cost, please see our general response to “training and inference cost should be provided / lack of discussion about computational cost”.
>
> \
> > Ablation clarity: more detailed ablation studies would help clarify the contribution of each component, particularly the individual modules within ReSIM and Dyna-GRPO
>
> Thanks for the question. Please refer to our general response to “Ablation analysis is incomplete / more detailed ablation studies would help”.

---

> ### Author Response · Authors · 2025-11-17
> **Response to Questions**
>
> > About $s^{refine}_t$, since it consists of state-action pairs, could its length pose an issue for the LLM input? Specifically, is there a risk that the sequence could become too long for the model to handle efficiently? Is the model performance sensitive to d?
>
>
> Thank you for this question. During Dynn-GRPO training we did set an upper limit of d=5 to prevent potential length issues. We note that d is determined by the model, since it is the depth of the plan that it generated at each step (capped at 5). So we do not think it is “sensitive to d”, as it is mostly controlled by the model itself.
>
> We have added this in our Section 3.3 in our revision.

---

> > ### Comment · Reviewer_J1B4 · 2025-11-26
> > **Reviewer response**
> >
> > Thanks to the authors for their response. My concerns have been addressed. I think the paper is suitable for a poster acceptance, and I maintain my positive score.

---

### Official Review · Reviewer_K5cm · 2025-10-28

**Soundness:** 3
**Presentation:** 3
**Contribution:** 3
**Rating:** 6
**Confidence:** 1

**Summary:**

The authors present Dyna-Mind, a training framework that teaches a VLM to learn a simulation of the underlying environment and incorporate it into learning. A two-stage training framework is presented, including a method for training agents to produce reasoning traces called Reasoning with Simulations (ReSim), and an online reinforcement learning algorithm called Dyna-GRPO.

**Strengths:**

The work is decently situated in the field. The math is extensive. Dyna-GRPO is an interesting implementation of model-based RL. Results are clear and well presented. A relatively large number of environments are used.

**Weaknesses:**

The paper is slightly confusing at times, there is slight misuse of the notion of “policy” (see comments below).

**Questions:**

There is this notion of the “simulation ability of the policy” and it’s not clear what that means. Typically in RL the policy is not simulating any part of the environment, it only outputs an action.

It would be useful to make an explicit distinction between the environment model and the policy, just as in Dyna-Q. It seems that the LLM is generating entire rollouts, including actions and states? So the environment model and policy are represented by a single LLM?

---

> ### Author Response · Authors · 2025-11-17
> **Response to Weaknesses and Questions**
>
> We thank reviewer K5cm for acknowledging our method as an “interesting” instantiation of Dyna and “clear and well presented” result!
>
> ---
>
> > The paper is slightly confusing at times, there is slight misuse of the notion of “policy”. There is this notion of the “simulation ability of the policy” and it’s not clear what that means. Typically in RL the policy is not simulating any part of the environment, it only outputs an action… It seems that the LLM is generating entire rollouts, including actions and states? So the environment model and policy are represented by a single LLM?
>
>
> Yes, our trained model is able to generate rollouts during its reasoning process, effectively modeling both the environment and the policy. This is the core novelty and motivation of our work: to synergize simulation (modeling future states) and acting (determining optimal actions) within the *reasoning* of a single (V)LM.
>
> For example, Table A4 demonstrates a reasoning trace that simulates three potential rollouts before the model finally selects “go to diningtable 1” as the next action.

---

### Official Review · Reviewer_Tcgq · 2025-10-28

**Soundness:** 3
**Presentation:** 3
**Contribution:** 3
**Rating:** 6
**Confidence:** 4

**Summary:**

This paper introduces Dyna-Mind, a two-stage training framework for teaching (V)LM agents to simulate environment dynamics during reasoning.
In Stage 1 (RESIM), the model learns to generate reasoning traces grounded in real environment rollouts by converting search trees into structured simulations.
In Stage 2 (Dyna-GRPO), an online reinforcement learning phase further improves both simulation and policy quality using feedback from intermediate and final outcomes.
Experiments on Sokoban, ALFWorld, and AndroidWorld show that Dyna-Mind enhances long-horizon planning and reasoning compared with existing baselines.

**Strengths:**

1. Presents a clear two-stage design that mirrors the human cognitive process of “mental simulation before action.”
2. RESIM effectively links reasoning traces with real-world dynamics, providing grounded supervision.
3. Dyna-GRPO adds an elegant reinforcement step that combines intermediate and outcome feedback.
4. Evaluation covers synthetic-to-realistic benchmarks and uses diverse metrics (success rate, simulation quality, token cost).
5. Writing and motivation are strong; results convincingly support the claimed improvements.

**Weaknesses:**

1. Ablation analysis is incomplete: it is unclear how much gain comes from (a) using real future states vs. (b) stronger aggregation LLMs or A-refine components.
2. The RESIM data-selection process for choosing “best plans” is under-specified, criteria for simulation quality and filtering are not explained.
3. Minor presentation issues: inconsistent decimals in tables, unclear “1.0×” unit in Table 1, and a few typos (e.g., planing-intensive 2 planning-intensive).
4. Lack of discussion about computational cost and scalability of RESIM when applied to larger real-world environments.

**Questions:**

The paper proposes a well-motivated and technically coherent framework that bridges reasoning and simulation. Although ablation and data-quality analyses are missing, the idea is novel and the results are promising for advancing long-horizon agent research. Moreover, pleaser refer to the weakness.

---

> ### Author Response · Authors · 2025-11-17
> **Response to Weaknesses**
>
> We thank reviewer Tcgq for acknowledging our “elegant” method design, clear motivation and writing, and strong results across diverse benchmarks (Sokoban, ALFWorld, and AndroidWorld)!
>
> ---
>
> > Ablation analysis is incomplete: it is unclear how much gain comes from (a) using real future states vs. (b) stronger aggregation LLMs or A-refine components.
>
>
> For (b), we believe that using LLMs stronger than DeepSeek-V3 (685B) should not be necessary. For ReSIM, the model is only tasked with summarizing the existing search tree branches into a response, a task for which a strong enough LLM, such as DeepSeek-V3, should be sufficient. To demonstrate this, we provide results on using Qwen3-235B-A22B-Instruct for aggregation below, which showed a similar performance.
>
> | Method | Sokoban |
> | :--- | :--- |
> | Distill(ReSIM) w/ DeepSeek-V3 | 71.9±1.5  |
> | Distill(ReSIM) w/ Qwen3-235B-A22B | 71.0±1.6 |
>
> For (a), we are unsure what could be an alternative. Maybe you mean using "simulated" future states generated by an LLM? We believe this is highly likely to underperform compared to using real future states, as LLM-generated information is often factually incorrect.
>
> Last but not least, for a more complete ablation study, please refer to our general response to “Ablation analysis is incomplete / more detailed ablation studies would help”.
>
> \
> > Minor presentation issues: inconsistent decimals in tables, unclear “1.0×” unit in Table 1, and a few typos (e.g., planing-intensive 2 planning-intensive).
>
> In Table 1, we measured the “Gen. Token” *relative* to that of ReACT(Qwen2.5-7B-Instruct), as we believe this makes comparisons easier. We added this clarification in our Table 1 and Table 3 in our revision.
>
> Thanks for finding the typo! We have fixed this in our revision.
>
>
> \
> > Lack of discussion about computational cost and scalability of RESIM when applied to larger real-world environments.
>
> Regarding computation cost, please see our general response to “training and inference cost should be provided / lack of discussion about computational cost”.
>
> Regarding a “larger real-world environment”, we note that in this work we evaluated not only on Sokoban and ALFWorld but ***also on AndroidWorld***, which is a highly complex real-world environment about controlling realistic android devices (see Section 4.2). We believe our strong results in all three benchmarks demonstrates the effectiveness of our work.

---

> > ### Comment · Reviewer_Tcgq · 2025-11-24
> >
> > The rebuttal does not address my original concern, which was about compute and scalability, not benchmark diversity. To assess whether RESIM is practically usable and reproducible, the paper should report at least high-level compute statistics (hardware, training GPU hours, approximate FLOPs per step, and inference cost vs. baselines). Without such information, it remains unclear whether the claimed efficiency holds in realistic settings or can be replicated by others.

---

> ### Author Response · Authors · 2025-11-24
>
> > The rebuttal does not address my original concern… To assess whether RESIM is practically usable and reproducible, the paper should report at least high-level compute statistics (hardware, training GPU hours, approximate FLOPs per step, and inference cost vs. baselines).
>
>
>
> In our general response, we noted that training data for baselines was generated using compute comparable to ReSIM, by choosing best-of-N sampling such that N approximately matches ReSIM’s compute. For clarity, we provide detailed statistics for ALFWorld below. We observe that our Distill(ReSIM) uses a similar data generation compute as Distill(V3) (by construction), **while requiring less training compute and achieving substantially higher performance than both Distill(V3) and Distill(R1).**
>
>
> | Method | Data Gen. Tokens (10^3) | Training FLOPs (10^12) | Inference Gen. Tokens | Performance |
> | :-- | :-- | :-- | :-- | :-- |
> | Distill(V3) | 5.9 | 112 | 367 | 57.8 |
> | Distill(R1) | 40.8 | 199 | 2552 | 56.8  |
> | Distill(ReSIM) | 6.1 | 47 | 341 | 74.1 |
>
>
> where “Data Gen. Tokens” denotes the average tokens used per turn to construct training trajectories (**for Distill(ReSIM), this corresponds to ReSIM’s compute**); “Training FLOPs” is the total SFT compute for distillation; and “Inference Gen. Tokens” is the average tokens generated per turn during evaluation.
>
>
> We have added this table in our general response. Please let us know if you have any further concerns.

---

> > ### Comment · Reviewer_Tcgq · 2025-11-25
> >
> > Thank you very much for the clarifications. I will keep my positive scores. Best of luck!

---

### Official Review · Reviewer_z3zA · 2025-10-29

**Soundness:** 3
**Presentation:** 2
**Contribution:** 3
**Rating:** 4
**Confidence:** 3

**Summary:**

This paper aims to enable a model to make decisions directly through internal simulation during inference, rather than by directly experiencing each option. The authors propose Dyna-Mind, a two-stage framework to achieve this goal. In the first stage, a search tree is constructed to build reasoning trajectories, and a VLM is trained to distill actions based on generated trajectories. In the second stage, reinforcement learning further refines both the decision-making strategy and the model’s simulation capability. Experimental results show that Dyna-Mind outperforms weaker LLMs and previous approaches, including Dyna-Think, and surpasses DeepSeek-R1 in more complex environments.

**Strengths:**

1. The paper presents a novel idea that enables the model to simulate the environment during reasoning rather than performing actual physical simulations.
2. The paper achieves better results not only on Sokoban but also on the other complex environments compared to other LLM methods.

**Weaknesses:**

1. This paper presents a comprehensive algorithm with several new proposed components, including ReSim, ReSim distillation, and Dyna-GRPO. However, it is not clear what the motivation is for the design of these components. Overall, the proposed architecture is a bit complicated without clear explanations. For example, ReSim first proposes a tree search-based method to simulate and generate trajectories. However, in Dyna-GRPO, there is another SimRollout approach to do the same thing. In addition, it is also unclear to use three different types of trajectories. Including an ablation study for each component would be more convincing.

2. In the experiments, comparisons should also be made with other search-based methods mentioned in the Related Work section, such as ExACT.

3. The training process requires several steps, including constructing tree search reasoning in ReSim and generating Rollout/SimRollout in Dyna-GRPO, which increases the training cost. The paper should provide the training computational cost compared to other methods. In addition, the inference computational cost should also be provided for a more complete evaluation.

**Questions:**

1. Please address the concerns raised in the weaknesses.

2. The definition of $a^{refine}$ in Line 274 is not sufficiently clear, and it is uncertain whether it is identical to the definition given in Line 275. Do the authors first generate $\tau'_{\text{refine}}$ and then remove the future-state information from it to obtain $\tau'$?

3. In Algorithm 1, the method requires different kinds of rollouts, such as $\tau$, $\tau'$, and $\tau'_{\text{refine}}$​, but without providing an ablation study for using different rollouts in the training. What would be the impact on performance if one of them were removed?

4. What is the reason for the use of distillation? In Table 1, it seems that both ReAct and RESIM can also be evaluated on the OOD test set. What is the reason for not providing these results?

5. In Table 1, what would be the performance if RESIM (without distill) were combined with Dyna-GRPO? Or the Distill(R1)+DYNA-GRPO?

6. In the AndroidWorld experiment analysis, the paper attributes the suboptimal performance to the rollout model (Qwen2.5-VL-72B). It would be beneficial to provide more concrete evidence supporting this claim or modify the method to demonstrate this limitation better.

---

> ### Author Response · Authors · 2025-11-17
> **Response to Weaknesses**
>
> We thank reviewer z3za for acknowledging our novelty and strong results across diverse benchmarks (Sokoban, ALFWorld, and AndroidWorld)!
>
> ---
>
> > This paper presents a comprehensive algorithm with several new proposed components, including ReSim, ReSim distillation, and Dyna-GRPO. However, it is not clear what the motivation is for the design of these components.
>
> ReSIM’s goal is to enable the base model to simulate during its reasoning (note that models such as Qwen-2.5-7B cannot reason with world simulation without training). Therefore, ReSIM first constructs imitation learning data where the reasoning process consists of explicitly planning with simulations. Then, ReSIM distillation simply performs SFT to train the model with this simulation-guided reasoning.
>
> While ReSIM provides a principled way to synergize simulation with reasoning, it is relied on multiple modules to construct training data. Additionally, such offline training may limit models’ generalization ability to new tasks. To address this, we propose DYNA-GRPO, a modification of GRPO to further improve the model's simulation ability (i.e., via SimRollout) during online RL.
>
>
> Overall, both ReSIM and Dyna-GRPO aim to enhance reasoning by improving the model's simulation ability. For a quantitative analysis of this simulation ability, we refer to Table 2, which shows that our methods effectively improved the both simulation score and overall task performance.
>
> \
> > In the experiments, comparisons should also be made with other search-based methods mentioned in the Related Work section, such as ExACT.
>
> We note that ReSIM is not a new search algorithm, but rather a new method to *utilize the search trees generated by (existing) search algorithms* to construct simulation-guided reasoning traces. Once the model is trained (e.g. via ReSIM/Dyna-GRPO), *no search algorithm is used during inference*. Therefore, we believe search-based algorithms are not comparable to ReSIM, as they are not intended for training.
>
> In general, we do believe that using more advanced search algorithms such as MCTS, R-MCTS (i.e., proposed by ExACT), ToT, and more on L195-197 and L93-94 could further improve our ReSIM result, but we highlight that with a simple DFS we already outperformed models such as DeepSeek-R1, as shown in Table 1. To disentangle what search algorithm is used v.s. ReSIM training performance, we present out ReSIM(BFS) result below:
>
>
> | Method | Sokoban | ALFWorld |
> | :--- | :---  | :---  |
> | ReSIM(DFS) | 71.9% | 78.9% |
> | ReSIM(BFS) | 71.1% | 78.9% |
>
> This result suggests that performance of ReSIM does not specifically rely on DFS. We added this discussion in our revised manuscript.
>
> \
> > The training process requires several steps, including constructing tree search reasoning in ReSim and generating Rollout/SimRollout in Dyna-GRPO, which increases the training cost. The paper should provide the training computational cost compared to other methods. In addition, the inference computational cost should also be provided for a more complete evaluation.
>
> Please see our general response to “training and inference cost should be provided / lack of discussion about computational cost”. We note that during inference, all trained models does *not* use any search algorithm and simply used the same prompt as in ReACT(...) baselines.

---

> ### Author Response · Authors · 2025-11-17
> **Response to Questions**
>
> > The definition of $a^{refine}$ in Line 274 is not sufficiently clear, and it is uncertain whether it is identical to the definition given in Line 275. Do the authors first generate $\tau^{refine}$ and then remove the future-state information from it to obtain $\tau^{‘}$?
>
>
> Yes, $a^{refine}$ is the same in both $\tau^{refine}$ and $\tau^{\prime}$, and in practice we simply first generated $\tau^{refine}$ and then removed the future-state information to obtain $\tau^{\prime}$ as you said.
>
> Regarding the definition of $a^{refine}$, we explained this in L256-257 and provided an illustration in Figure 4. We are happy to elaborate more if any specific aspect of L256-257/Figure 4 remains unclear.
>
> \
> > In Algorithm 1, the method requires different kinds of rollouts but without providing an ablation study for using different rollouts in the training. What would be the impact on performance if one of them were removed?
>
> We note that both $\tau^{refine}$ and $\tau^{\prime}$ are computed simultaneously using SimRollout, where we simply first generated $\tau^{refine}$ and then removed the future-state information to obtain $\tau^{\prime}$ (see previous response). If we remove $\tau^{refine}$ and $\tau^{\prime}$, the algorithm reduces to GRPO. Therefore, we ablated the effect of $A_{refine}$ which uses $\tau^{refine}$. Please see the results below. For a more detailed ablation result, please see our general response to “Ablation analysis is incomplete / more detailed ablation studies would help”.
>
> | Method | Sokoban | ALFWorld |
> | :--- | :--- | :--- |
> | Dyna-GRPO | 82.5% | 92.5% |
> | - $A_{refine}$ | 80.8% | 89.2% |
> | - SimRollout | 79.1% | 87.0% |
>
>
> \
> > In Table 1, it seems that both ReAct and RESIM can also be evaluated on the OOD test set. What is the reason for not providing these results?
>
>
> The distinction between ID and OOD is only meaningful for finetuned models. These baselines (e.g., ReACT(Qwen2.5-7B-Instruct), ReACT(GPT-4o), ReACT(Claude-3.7-Sonnet), etc.) are *not* finetuned. Consequently, both the ID and OOD test sets are effectively "OOD" for these models. We therefore reported their performance under the "ID" column for simplicity.
>
> \
> > In the AndroidWorld experiment analysis, the paper attributes the suboptimal performance to the rollout model (Qwen2.5-VL-72B). It would be beneficial to provide more concrete evidence supporting this claim or modify the method to demonstrate this limitation better.
>
> We believe this can be seen from the performance of the rollout policy model. As shown in Table 3, the rollout policy (Qwen-72B) achieves only 19.5±0.4. It is unlikely that a simple search algorithm like DFS could bridge this significant gap to the high performance seen in other tasks (e.g., Qwen-32B already achieved 36.7 and 36.2 in Sokoban and ALFWorld, respectively).
>
> In general, we believe most search-related methods are first bounded by the base rollout model's performance: if no correct action is sampled within the tree, the task cannot be solved [1, 2].
>
> ---
>
> Reference
>
>
> [1] Yu, Xiao, et al. "Exact: Teaching ai agents to explore with reflective-mcts and exploratory learning." arXiv preprint arXiv:2410.02052 (2024).
>
> [2] Snell, Charlie, et al. "Scaling llm test-time compute optimally can be more effective than scaling model parameters." arXiv preprint arXiv:2408.03314 (2024).

---

> > ### Comment · Reviewer_z3zA · 2025-11-26
> >
> > Thank you for the detailed clarifications and the additional experiments. The revisions improve the clarity and address some of my concerns. However, I still have two concerns.
> >
> > 1. For the ablation study, as some results are very close,  could you report the standard deviation for each row?
> >
> > 2. In addition, regarding training costs, only the costs for stage 1 are reported. However, for stage 2 (RL training), the proposed method requires additional cost for tree search reasoning. Although the authors mentioned "the increase in training time is small (<=25%)" but without reporting any numbers. In fact, 25% is still a non-trivial cost. Could you report the actual training time for stage 1 and stage 2 of each method in Table 1?

---

> ### Author Response · Authors · 2025-11-24
>
> Dear reviewer z3zA,
>
> Thank you again for these questions and feedback! Please let us know at your earliest convenience if you have any further questions or concerns, or would like to us conduct any additional experiments.

---

> ### Author Response · Authors · 2025-11-26
>
> > For the ablation study, as some results are very close, could you report the standard deviation for each row?
>
>
> We have added the standard deviations below. We observe that each row is on average >1 standard deviation away from its counterpart (except for "- DFS+BFS”). We note that “- DFS+BFS” is near identical to “Distill(ReSIM)”, but that is *desired* as it shows that ReSIM is not dependent on the specific search algorithm used.
>
> | Method | Sokoban | ALFWorld |
> | :--- | :--- | :--- |
> | Dyna-GRPO | 82.5%pm1.5 | 92.5%pm0.8 |
> | - $A_{refine}$ | 80.8%pm1.0 | 89.2%pm0.5 |
> | - SimRollout | 79.1%pm1.3 | 87.0%pm3.2 |
> | Distill(ReSIM) | 71.9%pm1.5 | 78.9%pm2.1 |
> | - DFS+BFS | 71.1%pm1.5 | 78.9%pm1.8 |
> | - DFS | 49.2%pm1.1 | 58.9%pm1.1 |
>
>
>
> > Although the authors mentioned "the increase in training time is small (<=25%)" but without reporting any numbers. In fact, 25% is still a non-trivial cost. Could you report the actual training time for stage 1 and stage 2 of each method in Table 1?
>
>
> We believe the “25%” we mentioned is itself a number. However, to be more precise we provide the following table to complement our results in overall response. We observe that although both GRPO and Dyna-GRPO (ours) increases 11% and 21% of the training time compared to RLOO, respectively, but 1) our method is more efficient as it achieves the best performance achievable by RLOO much faster; 2) our final model generates less token and achieves an overall higher performance. *Overall, we believe this increase in one-time training cost is significantly amortized by our method’s improved training efficiency and higher final performance.*
>
>
> | Method | Total Training Time | Time to Reach Best RLOO | Inference Gen. Tokens | Performance |
> | :-- | :-- | :-- | :-- | :-- |
> | Distill(ReSIM)+RLOO | 15.1H  | 15.1H | 447 | 85.9 |
> | Distill(ReSIM)+GRPO |  16.8H | 11.1H | 348 | 87.0 |
> | Distill(ReSIM)+Dyna-GRPO | 18.3H  | 7.2H | 262 | 92.5 |
>
> where “Time to Reach Best RLOO” is the training time to reach 85.9% (final performance achieved by RLOO).

---

### Author Response · Authors · 2025-11-17
**General Response**

We thank all reviewers for their helpful feedback and suggestions! We have also updated our paper accordingly. All changes are highlighted in red for ease of reference. Below we address some common questions/concerns.

---

> Ablation analysis is incomplete / more detailed ablation studies would help

We already included ablations for each component of Dyna-Mind in Table 1, presented in separate blocks. We have also added a few relevant items from our reviews to strengthen our results. Please see the table below.


| Method | Sokoban | ALFWorld |
| :--- | :--- | :--- |
| Dyna-GRPO | 82.5% | 92.5% |
| - $A_{refine}$ | 80.8% | 89.2% |
| - SimRollout | 79.1% | 87.0% |
| Distill(ReSIM) | 71.9% | 78.9% |
| - DFS+BFS | 71.1% | 78.9% |
| - DFS | 49.2% | 58.9% |

where “-DFS+BFS” indicates replacing the DFS used in ReSIM with BFS. We note that removing SimRollout from Dyna-GRPO simply reduces the algorithm to GRPO; and removing DFS from ReSIM reduces to Distill(V3). Both of these baselines are already included in our Table 1-3.

For clarity, we have added this table in our revised manuscript.


\

> training and inference cost should be provided / lack of discussion about computational cost

Regarding inference cost, we provided them for our ReSIM- and Dyna-GRPO-trained models in Table 1 (Sokoban, ALFWorld) and Table 3 (AndroidWorld). Our models demonstrate a *lower* inference cost than all baselines, achieving better performance while generating fewer tokens per response. This efficiency is detailed in the "Gen. Token" column, which is measured relative to the ReACT(Qwen2.5-7B-Instruct) baseline.

Regarding training cost, we ensured a fair comparison by constructing baselines (Distill(V3), Distill(R1)) using compute similar to that of ReSIM. We achieved this by using correct trajectories from best-of-N sampling, where N was set to approximately match the compute used in ReSIM, in addition to only training on tasks where *all* methods correctly solved (L348-349). For more detailed statistics, please see the table below on ALFWorld. For Dyna-GRPO, the increase in training time is small (<=25%), which contrasts favorably with recent algorithms like VinePPO [1] and TreeRL [2] that incur significant training overhead. Overall, we believe this increase in one-time training cost is significantly amortized by our model's higher performance and lower-cost inference.


| Method | Data Gen. Tokens (10^3) | Training FLOPs (10^12) | Inference Gen. Tokens | Performance |
| :-- | :-- | :-- | :-- | :-- |
| Distill(V3) | 5.9 | 112 | 367 | 57.8 |
| Distill(R1) | 40.8 | 199 | 2552 | 56.8  |
| Distill(ReSIM) | 6.1 | 47 | 341 | 74.1 |


where “Data Gen. Tokens” denotes the average tokens used per turn to construct training trajectories (for Distill(ReSIM), this corresponds to ReSIM’s compute); “Training FLOPs” is the total SFT compute for distillation; and “Inference Gen. Tokens” is the average tokens generated per turn during evaluation. We observe that our Distill(ReSIM) uses a similar data generation compute as Distill(V3) (by construction), *while requiring less training compute and achieving substantially higher performance than both Distill(V3) and Distill(R1).*


---

Reference

[1] Kazemnejad, Amirhossein, et al. "VinePPO: Refining Credit Assignment in RL Training of LLMs." arXiv preprint arXiv:2410.01679 (2024).

[2] Hou, Zhenyu, et al. "TreeRL: LLM Reinforcement Learning with On-Policy Tree Search." arXiv preprint arXiv:2506.11902 (2025).

---

### Author Response · Authors · 2025-12-03

Dear AC, SAC, PC, and reviewers,

We would like to express our sincere gratitude for the time and effort devoted to the review and the discussion period. We are also saddened that this review cycle was affected by the recent unexpected event. Below, we provide a concise summary of the main questions and concerns raised during the rebuttal period and how our responses addressed them. We have also updated the manuscript accordingly, with revisions marked in red.


More specifically:
- **Reviewer z3zA** acknowledges our method’s novelty and strong results across Sokoban, ALFWorld, and AndroidWorld. The reviewer first noted that “it is not clear what the motivation is” for our methods, to which we provided additional explanation. The reviewer also suggested comparing against “other search-based methods.” We responded with results showing that BFS achieves similar performance, while emphasizing that ReSIM is *not a new search algorithm* but a method for leveraging search trees to construct simulation-guided reasoning traces. In the follow-up response, reviewer z3zA *confirmed that our responses and revisions addressed these questions/concerns above*. The reviewer then further requested (1) standard deviations for the ablation studies. We provided a table showing that each row is, on average, >1 standard deviation apart from its counterpart. (2) Provide the “actual training time for stage 2” in addition to stage 1 (which we provided in the general response). We supplied a detailed runtime breakdown for our stage 2 training, demonstrating that our method achieves better performance while being more efficient (achieving the best performance achievable by RLOO much faster than GRPO).
- **Reviewer Tcgq** acknowledges our “elegant” method design, clear motivation and writing, and strong results across diverse benchmarks (Sokoban, ALFWorld, and AndroidWorld). The reviewer raised concerns about (1) “Ablation analysis is incomplete”, where we responded with a more complete ablation analysis in our general response; (2) “Minor presentation issues”, where we updated our manuscript accordingly; and (3) “Lack of discussion about computational cost”, where we additionally provided a detailed computation cost breakdown and showed that our Distill(ReSIM) uses a similar data generation compute as Distill(V3), while requiring less training compute and achieving substantially higher performance than both our baselines. In the subsequent response, the reviewer *acknowledges that our responses addressed all questions and concerns, and decided to keep the positive score*.
- **Reviewer K5cm** acknowledges our method as an “interesting” instantiation of Dyna and “clear and well presented” result. The reviewer notes that “The paper is slightly confusing at times… It seems that the LLM is generating entire rollouts, including actions and states? So the environment model and policy are represented by a single LLM?”, where we responded by explaining that our trained model is indeed able to generate rollouts during its reasoning process, effectively modeling both the environment and the policy. We also pointed to Table A4 as an example, which showed a reasoning trace that simulates three potential rollouts before the model finally selects “go to diningtable 1” as the next action.
- **Reviewer J1B4** acknowledges our “novel conceptual contribution”, “clear and systematic” method, and strong and diverse results across three benchmarks (Sokoban, ALFWorld, and AndroidWorld). The reviewer first raises the concern about “High computation cost … it remains unclear whether the method can scale efficiently to large environments or applications,” where we responded that (1) in Section 4.2 we also tested on AndroidWorld, which is a highly complex and large scale environment about controlling android devices; and (2) we additionally provided a detailed computation cost breakdown in our general response. The reviewer also requested for a “more detailed ablation study”, which we provided in our general response. In the subsequent response, reviewer J1B4 *acknowledges that our responses addressed all questions and concerns, decided to keep the positive score, and “think the paper is suitable for a poster acceptance”*.



Overall, we believe our responses address the core concerns and clarification questions raised by all reviewers. We hope the summary above helps the AC, SAC, and PCs more easily digest the key outcomes of the discussion period.


Sincerely, The Authors

---

### Meta-Review · Area_Chair_vyHp · 2025-12-22

**Summary:**

### Summary
This paper introduces Dyna-Mind, a two-stage training framework that teaches VLM agents to mentally simulate alternative futures before acting, via ReSim and Dyna-GRPO.

### Reviewer summary
Reviewers highlight the cognitive-inspired mental simulation before action motivation and find the two-stage design coherent and elegant. The reported results are strong. However, the ablations initially insufficient, requests for training & inference cost statistics and clearer cost comparisons to baselines.

### AC Comments

From AC's view, the core idea is compelling and timely for long-horizon agent research, and the results are consistently positive across three benchmarks including a realistic setting. Crucially, the rebuttal and follow-up discussion appear to substantively address the main reviewer concerns. Given the overall reviewer sentiment and that key issues were resolved in discussion, I believe the paper meets the bar for acceptance, with remaining limitations primarily around broader scaling/generalization rather than core validity.

**Reviewer Concerns:**

### Reviewer z3zA

* Motivation / complexity / ablations; comparisons to search-based methods; cost reporting

largely resolved via added clarifications, extra ablations (including BFS vs DFS), std dev reporting, and explicit stage-1 vs stage-2 training time breakdown after follow-up.
residual still somewhat complex pipeline, but the added quantitative breakdowns substantially improve clarity and evaluation completeness.


### Reviewer Tcgq

* Compute/scalability concern

Resolved after authors provided high-level compute statistics and the reviewer explicitly acknowledged the clarification and maintained a positive score.

* Ablation/data-selection specification & minor presentation issues

Resolved / partially resolved via additional ablations and manuscript fixes.

### Reviewer K5cm

* Conceptual confusion (“policy” vs environment model; simulation ability)

Resolved by the authors’ clarification that the trained model simulates rollouts during reasoning (jointly modeling dynamics + action choice), with examples referenced. Confidence remains low due to expertise mismatch, but the concern itself is addressed.

### Reviewer J1B4

* Compute cost / scalability & ablation clarity

Resolved after cost breakdown + expanded ablation response; reviewer explicitly states concerns were addressed and the paper is suitable for poster acceptance, maintaining the positive score.

**Reviewer Scores:**

* z3zA (4): likely increase to 6 after the authors provided the requested standard deviations and explicit stage-2 training time numbers (the two remaining concerns).

* Tcgq (6): unchanged; the reviewer explicitly acknowledges the compute clarifications and maintains the positive score.

* K5cm (6,): likely unchange, since the main confusion was clarified but the reviewer self-reported limited expertise.

* J1B4 (6): unchanged; reviewer states concerns are addressed and supports poster acceptance.

---

### Decision · Program_Chairs · 2026-01-26

Accept (Poster)